# An ecological approach to structural flexibility in online communication systems

María J. Palazzi [1], Albert Solé-Ribalta[1,2], Violeta Calleja-Solanas [3], Sandro Meloni[3], Carlos A. Plata [4,5], Samir Suweis [4] & Javier Borge-Holthoefer [1✉]

Human cognitive abilities are limited resources. Today, in the age of cheap information—cheap to produce, to manipulate, to disseminate—this cognitive bottleneck translates into hypercompetition for rewarding outcomes among actors. These incentives push actors to mutualistically interact with specific memes, seeking the virality of their messages. In turn, memes' chances to persist and spread are subject to changes in the communication environment. In spite of all this complexity, here we show that the underlying architecture of empirical actor-meme information ecosystems evolves into recurring emergent patterns. We then propose an ecology-inspired modelling framework, bringing to light the precise mechanisms causing the observed flexible structural reorganisation. The model predicts—and the data confirm—that users' struggle for visibility induces a re-equilibration of the network's mesoscale towards self-similar nested arrangements. Our final microscale insights suggest that flexibility at the structural level is not mirrored at the dynamical one.

[1] Internet Interdisciplinary Institute (IN3), Universitat Oberta de Catalunya, Barcelona, Catalonia, Spain. [2] URPP Social Networks, University of Zurich, Zurich, Switzerland. [3] IFISC, Institute for Cross-Disciplinary Physics and Complex Systems (CSIC-UIB), Palma de Mallorca, Spain. [4] Dipartimento di Fisica e Astronomia G. Galilei, Università di Padova, Padova, Italy. [5] Université Paris-Saclay, CNRS, LPTMS, Orsay, France. ✉email: jborgeh@uoc.edu

Our current experience of the accelerated stream of digital content[1] has exposed, in full range, the tight bio-cognitive limitations that we are subject to[2,3]. Their finiteness had not, in general, arisen in everyday communication processes: not in the pre-industrial age, where a physical (face to face) or low-bandwidth interaction governed the slow change of public opinion; nor during the predominance of mass media, when the exposure to an oligopolistic media environment put little pressure to the attentional resources of the audience. In both cases, the public sphere was hierarchically structured and framed by the operations of few actors on a rather slow time scale. Contrarily, the paradigm of online communication is characterised by the fragmentation of the public sphere[4], in which elite and non-elite actors behave like information sources and receivers on the virtual stage. Only in this new scenario, attention, memory and processing time, the cognitive underpinnings of visibility[5–7], suddenly become critical assets to compete for[8–13]: their scarcity has been exposed.

On the other hand, the idea that words (or, more generally, memes) compete to be used by speakers is one of the fundamentals of cultural evolution, a dominant framework to explain language change in the last three decades[14–16]. Although the accounts within this discipline differ in their theoretical background and assumptions, they share the postulate that linguistic units are 'replicators': their survival depends on their ability to be copied, i.e., internalised by other speakers of the language. Lexical theorists argue that the scarce resource which memes compete for is the speakers' attention (users, in the online context). Words, in this context, evolve to become compressible (reduction or simplification) and discriminative (maintaining relevant distinctions). In either case, the underlying assumption is that the cognitive system "wishes for" easier-to-internalise terms. While such theories (which precede the Internet) are not meant to explain lexical competition in online environments, the logic underlying these ideas hold as well[1].

Of course, the choice of a meme is context-dependent (past performance is no guarantee of future results), and thus the interactions between actors and memes are adaptive and extremely sensitive to changes in the communication environment—breaking news, fads and rumours, celebrity gatherings, etc. In turn, changes in the surrounding conditions tend to be ephemeral although frequent[1], in the more open and fluid access to many digital sources.

Complementary to direct competition (among actors and among memes), interaction across classes in the system can be thought of as mutualistic. In this work, we hypothesise that the existence of actor-meme mutualistic links is implied by the existence of the competitive ones: in a communication environment, the best tool for an actor to compete with its peers is precisely to build the best possible discourse by making the right lexical choices. These "information chunks" may—if correctly chosen—optimally spread information and consolidate the visibility they strive for[5–7]. Hence, for example, the (ab)use of hyper-emotional language that we suffer in nowadays politics[17], as an arms race to impact optimisation. On the other hand, the best way for a meme to compete with other lexical candidates is to be clear and concise: thus the mutualistic relationship—which does not exclude other strategies to maximise visibility.

Under the light of these four drivers—competition, mutualism, adaptation, environment—online communication systems and natural mutualistic assemblages can be understood as special cases of a broader class of mutualistic bipartite systems, i.e., those dominated by intra-class competition and inter-class mutually beneficial interactions. Although clearly functioning at very different spatial and temporal scales, this work evaluates, empirically and theoretically, whether this hypothesis holds.

Our failure to explore this possibility in the past is due to several factors. Previous approaches to an "info-ecological"

understanding of online communication dynamics typically focused on one of the dimensions of the problem (actors[8,10–12] or memes[1]), missing the structure-dynamics interplay of topologies and states in the network[18–20]. This picture changes dramatically if the focus is shifted from the relatively stable peer-to-peer network to the fluid information bipartite network, that is, ad hoc groups of users, which loosely gather around and engage in shared memes[21], operating in a hyper-competitive environment[17]. Other approaches, which did include the bipartite perspective, were limited to a qualitative discussion as a result of empirical observations on a single dataset[22], failing to identify the mechanisms that drive the whole system.

A picture that embraces the mentioned ingredients might open new promising possibilities to analyse and model online social networks if we consider that Ecology is rich in theoretical frameworks where the structure-dynamics coupling is studied[23–25]. Moreover, while testing these theories empirically in natural ecosystems is difficult—mainly because of the resource-intensive demands to collect data[26]—digital streams from social interactions are abundant on several spatial and temporal scales, and precise knowledge about the environmental (external) conditions —related to specific information flows—can also be collected.

The first problem to address under this information ecosystems framework is the network's structural volatility, which is coupled to the fluctuating nature of the environment. Online communication is heavily driven by the events surrounding it, which constantly trigger attention shifts that modify the behaviour of otherwise loosely linked assemblages of individuals and groups[17]. It is precisely this hectic, information-dense environment that dictates the emergence and fall of ephemeral synchronised attention episodes, which translate into fast structural changes.

Here, we provide evidence that information ecosystems exhibit structural flexibility[27] to environmental changes, recovering their original architecture in the aftermath of an external event affecting it. To do so, we stick to the "classical" scientific cycle –observation, hypothesis, model, prediction. First, we report on theory-free, empirical observations of the characteristic dynamical re-organisation in communication networks, as they react to environmental "shocks". Analysing the response of the Twitter ecosystem to different types of external events, we quantify how collective attention episodes reshape the user-hashtag information network, from a modular[28–30] to a nested[31–33] architecture, and back. The emergence of these structural signatures is consistent across different topics and time scales. Next, we propose a theoretical framework that explains the emergence of the patterns observed in real data streams, as a result of an adaptive mechanism. The model builds on the idea that the actor-meme network structure is effectively driven by an optimisation process[23], aiming at the maximisation of visibility[5–7], and that the nature of the user-meme interactions is mutualistic, i.e., beneficial for both. Furthermore, through our modelling framework, we predict that the users' struggle for visibility in any context facilitates the emergence of nested self-similar arrangements: either mesoscale (in-block) nestedness[34–40] during the compartmentalised stages or macroscale nestedness in exceptional global attention episodes. Eventually, we present some results that link our observations with the model at the microscale, which suggest that environmental shocks may leave a lasting trace on the system's node dynamical states, despite the structural flexibility found at the macroscale and mesoscale. These predictions are supported by the data.

## Results

**Structural flexibility in information systems.** Biased as it may be[41], Twitter is a sensitive platform that mirrors, practically

without delay[42–44], exogenous events occurring in offline environments. In this sense, Twitter data constitute a rich stream, providing a public and machine-readable reflection of the real world.

Despite the highly fluctuating nature of this communication stream, some reliable patterns emerge from its activity. We analyse these streams in a longitudinal manner, as a series of snapshots from time-resolved activity. Each slice is represented as a bipartite network with a fixed number of the most active actors (users in this context; $N_U = 2000$), and the corresponding memes (hashtags in this context; $N_H$) created and/or cited by these users, in that slice. Note that the set of users and hashtags in a given slice differs, to some extent, from one to the next: see Supplementary Note 1, and particularly Supplementary Fig. 2. Such a sequence of networks is studied monitoring different structural arrangements that are relevant to the dynamical stage in which the system is. For now, we focus on two of them: modularity[28,29] ($Q$) and nestedness[31,45,46] ($\mathcal{N}$). Details on the construction of time-resolved networks and their structural analysis can be found in the "Methods" section below, and Supplementary Note 1.

High levels of modularity correspond to a fragmented attention scenario and can be considered as the resting state of the system. In this stage, users mostly focus on their own topics of interest, i.e., a certain subset of hashtags, facilitating the emergence of identifiable blocks. High values of nestedness, on the other hand, reflect an extraordinary (and, thus, ephemeral) stage in which the system self-organises to attend one or few topics. In these cases, the discussion revolves around a small set of generalist hashtags (hashtags used virtually by everybody) and users (highly active individuals participating in many facets of the discussion).

Figure 1, panels c and h, present the evolution of $Q$ and $\mathcal{N}$ on two different portions of Twitter activity. For example, Fig. 1c corresponds to a period of over 45 days around the local elections in Spain (April-May 2019). For this dataset, the evolution of $Q$ and $\mathcal{N}$ shows a strongly anti-correlated behaviour. This behaviour can be explained by the mutual structural constraints that these two arrangements impose on each other, i.e., the upper bound for the co-existence of nested and modular structures. This bound is $Q \leq 1 - \mathcal{N}$, which implies that extremely high $\mathcal{N}$ values are incompatible with the high values of $Q$[47]. Note however that this does not impede many other regimes: it is perfectly possible (and actually frequent) that both $Q$ and $\mathcal{N}$ are extremely low (e.g., in Erdös-Rényi networks), or that both have intermediate values (see Supplementary Note 3). In sum, the significant growth of nestedness, qualitative and quantitative[48,49] (see the discussion on quantative nestedness in Supplementary Note 1), is not caused by a decrease in the modularity of the system, but, on the contrary, tightly linked to external events: see for instance the sudden changes in the structure on specific dates, shadowed in grey in Fig. 1c (debate and polling day, respectively). These extraordinary events are accompanied, unsurprisingly, by an increased volume of messages (Fig. 1a) and connectance. Despite previous research[50], neither volume nor connectance can explain, per se, the rapid surge of nestedness: see the study on the effects of activity in Supplementary Note 3 for a thorough discussion. The figure, at the scale of days, is complemented with high-resolution monitoring of portions of these exceptional events (Fig. 1d, e), which confirm the general anti-correlated trend. Finally, the most outstanding feature highlighted Fig. 1c is the structural flexibility: no matter how abrupt and large the excursion to a nested arrangement is, the system bounces back to its "ground"—predominantly modular—state soon after when the interest in the breaking news fades out.

The overall observed behaviour in Fig. 1c is replicated across different types of events. Figure 1h shows an equivalent pattern for a completely different event. In this case, the dataset comprises the reaction after the Nepal earthquake in 2015[51], including a major aftershock on May 12th. Unlike a political debate or an election date, this example is inherently unexpected and unpredictable—an important fact, attending the taxonomy of collective attention described in Lehmann et al.[42]. As in Fig. 1c, the coarse grain scale of days and weeks in Fig. 1h is complemented with high-resolution monitoring of a portion of exceptional events (panel i). Four more datasets are analysed in Supplementary Note 1. In all of them, the previous observations hold: the anti-correlated evolution of $Q$ and $\mathcal{N}$ (Supplementary Table 2), the alternation of modular and nested arrangements (Supplementary Fig. 4), etc. Also, in-depth discussions on statistical significance and the role of connectance can be found in Supplementary Note 3.

These analyses suggest that there is a tight logic underlying the structural fluctuations of the information network: the level of fragmentation of collective attention maps onto specific network arrangements, and is independent of the particular contents of the data stream. Online activity on different topics translate to comparable changes in the resulting patterns, no matter the semantics of the underlying discussion. The observed differences in the emergence, magnitude and persistence of structural changes are directly related to the predictability, intensity and duration of the exogenous events (i.e., related to the environmental conditions), and therefore cannot be explained as intrinsic to the communication system itself. The question remains, however, how a networked system can fluctuate so fast between two states which have often been considered incompatible[52,53]. The key to this puzzle is in-block nestedness, a hybrid modular-nested architecture that bridges the apparent antagonism between nestedness and modularity[47].

**Theoretical framework**. To understand the mechanisms that govern the observed flexibility, and, at the same time, to solve the puzzle around the network's nested-modular oscillations, we propose a model founded on the ecological drivers introduced above: competition, mutualism, co-adaptation and environment. The model builds on the simple idea that the network architecture between users and hashtags is the result of several local optimisation processes, i.e., each individual's maximisation of visibility, and that such process operates on top of attentional dynamics. To do so, we generalise the ecological adaptive modelling proposed by Suweis et al.[23,25], in which the system's actors (plant and pollinator species) strive for larger individual abundance, rewiring their interactions accordingly.

The synthetic information network model comprises a total of $N$ interacting "species" or nodes ($N_U$ users and $N_H$ hashtags), in which population dynamics—where population here quantifies the visibility of the users and/or of the hashtags—is driven by interspecific mutualistic interactions, following a Lotka-Volterra dynamics with Holling-Type II functional response[23,54].

Each species has an associated niche[55] which, in the context of an information ecosystem, represents their topical domain (i.e., the topic to which a user attends preferentially, and, conversely, the semantic space where a hashtag belongs to). For the sake of simplicity, each species' niche is represented as a Gaussian distribution with a given standard deviation $\sigma$[25]. Both users and hashtags niches are anchored around $T$ different points in the range [0,1], to express different topic preferences (users), and semantic domain (hashtags). To model the inherent diversity of users and hashtags within their topic, their position over the line

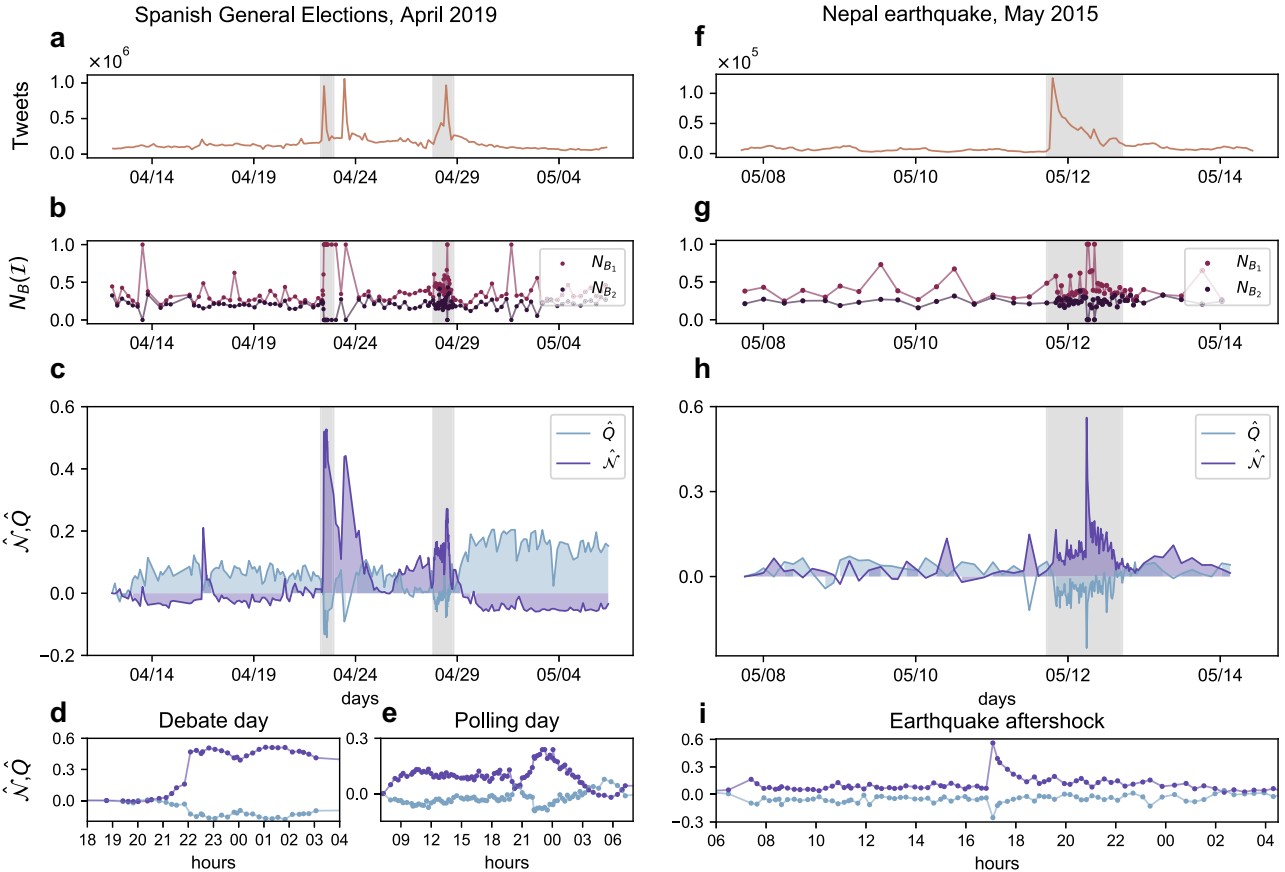

**Fig. 1 Structural measures over time for two datasets.** Here we show Twitter streams covering two different topics, i.e., Spanish general election of 2019 (panels **a**–**e**) and the 2015 Nepal earthquake (panel **f**–**i**), respectively. Spanning different time ranges and attracting varying levels of attention (see tweet volume in panels **a** and **f**), the information ecosystems self-organise in similar ways: a block organisation dominates the system (positive modularity $\hat{Q}$), reflecting the separate interests of users until external events induce large-scale attention shifts, which rearrange completely the network connectivity towards a nested architecture (high $\hat{\mathcal{N}}$, **c** and **h**). Note that, despite the predictable (Spain) vs. unpredictable (Nepal) nature of each stream, structural properties of the user-hashtag interaction networks ($\hat{Q}$ with $\hat{\mathcal{N}}$) are anti-correlated[22]. For a closer view, we highlight specific time windows in each dataset with some identifiable events happening in them (panels **d** and **e** for the Spanish elections dataset and panel **i** for the Nepal earthquake). In each plot, measures of modularity and nestedness are shifted from their initial values ($Q_0$ and $\mathcal{N}_0$, respectively). The panels **b** and **g**, corresponding to $N_B(\mathcal{I})$ highlight the nested self-similar arrangements at different scales, which is discussed later on.

is perturbed by a small amount, randomly sampled from a uniform distribution.

Competition occurs between species of the same class (or guild), whereas mutualistic interactions couple the dynamics of abundance of users and hashtags. Following the proposal of Cai et al.[25], the strength of the competitive interaction for attention[10,11,13] between a pair of users is tuned by a fixed parameter ($\Omega_c$) scaled by a quantity that depends on the niche overlap $G_{ij}$ between them. The same applies to hashtags, which compete to get used against close alternatives[15,16]. Similarly, the strength of the mutualistic interactions between a pair user-hashtag results from a fixed parameter ($\Omega_m$) scaled by the niche overlap between the pair user-hashtag—i.e., the similarity between the user's topic preference and the adequacy of the hashtag within this topic—and constrained to the existence of a link between them. Figure 2a summarises the ingredients of the model. We note that, in contrast to natural ecosystems, hashtags are an infinite resource—which explains why user-user competition does not grow with the number of shared hashtags.

On the dynamic side, each user attempts to change its mutualistic partners (hashtags) in order to maximise the benefit obtained from their use (see "Methods" section below, and Supplementary Note 2). This optimisation principle may then be

interpreted within an adaptive framework, in which users incrementally enhance their visibility by choosing the appropriate hashtags, see Fig. 2b. In this setting, only users can actively rewire their links to new hashtags. Within the model, this translates into reiterative rewiring interactions of randomly drawn users so as to increase their visibility—"abundance" in the ecological jargon.

Since our primary objective is to reproduce structural changes under the irruption of external events, the dynamical model includes as well a mechanism to introduce exogenous events in the environment. These can be understood as transitory shifts in the users' attentional niches, which are tantamount to (typically short-lived) changes in their interests (Fig. 2c). In this altered environment, users temporarily engage with new kinds of hashtags, different from those they usually interact with (see "Methods" section and Supplementary Note 2.

In an unperturbed simulated environment, the observed emergent structural arrangement mimics the prescribed organisation of niches in topical blocks. That is, a modular architecture arises from the random initial one, see Fig. 3e and f, for $t < 3 \times 10^4$ (note that the plot is shifted by $Q_0$, i.e., modularity at time $2 \times 10^3$, once the system has stabilised its architecture). This is in line with the resting state observed in the datasets (Fig. 1e, f), where users are focused on their own topics of interest. It is important to

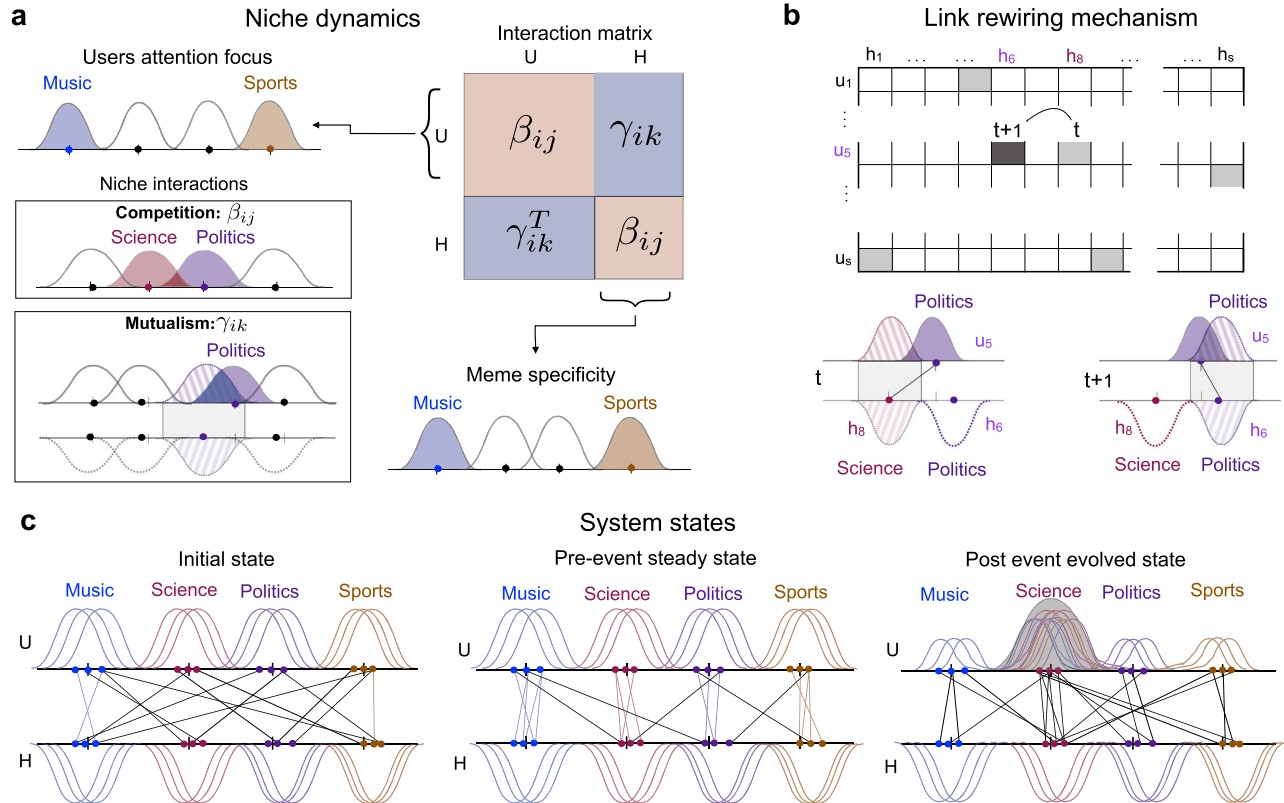

**Fig. 2 Schematic representation of the visibility optimisation model. a** Users and hashtags are represented as points in the range $[0, 1]$ in a niche axis. We modelled each niche as a Gaussian curve with a standard deviation $\sigma$. Topics are modelled as clusters of users (hashtags), i.e., $T = 4$. The coupling matrices $\beta$ and $\gamma$, which define the competitive (within guilds) and mutualistic interactions, are defined to be proportional to the niche overlap between pair of species. **b** At each time-step, species rewire their connections trying to optimise their abundance (popularity). If the rewiring leads to larger popularity the connection is kept, otherwise, the change is reverted. **c** Initially, the interactions are laid at random, and the rewiring process takes place. When the system reaches an evolved steady state, an external event enters the system. Users' niches are temporarily focused on a single common topic and the rewiring process continues while the effect of the event decays over time. As the event fades out, all species return to their original niche.

underline that the emergence of a modular architecture is not an artefact of the model: users (hashtags) do not rewire by similarity reasons; it is the search for an improvement in their individual visibility that naturally drives to the consolidation of those new connections. Also note that, in empirical settings, the random initial stage is impossible to observe since the network already has a modular organisation from the very beginning.

A change in the environment—e.g., breaking news—totally alters this scenario. The systems react with a decrease in $Q$, and an increase in the amount of nestedness in the system, Figure 3e, f for $3 \times 10^4 < t \lesssim 4 \times 10^4$. If the simulation refers to an abrupt event (Fig. 3f), the decrease in $Q$ is sharp and almost immediate; if the simulation refers to a predictable event (Fig. 3e), the collapse of $Q$ is smoother, and the emergence of nestedness is slightly delayed. Indeed, in this situation, we recover the results in Suweis et al.[23]—the emergence of global nestedness—because the existence of attentional niches becomes irrelevant when all niches are equally centred, at least on the users' side. In this sense, our niche-based population dynamics (and that of Cai's et al.[25]) is a generalisation of Suweis and co-authors' model. As the environmental shock fades out, the network architecture tends to recover the general layout present before the event was introduced, see $t \gg 4 \times 10^4$. The flexibility of empirical information ecosystems is thus replicated here, including the anti-correlated behaviour of $Q$ and $\mathcal{N}$ (Supplementary Note 3), and explained as a consequence of the adaptation to contextual changes—while the species' local strategies remain constant.

**Nestedness reframed: multi-scale analysis**. Beyond the examination of the evolution of $Q$ and $\mathcal{N}$, we now take a closer look at the intra-modular organisation of connections during the fragmentary stage of the system ($t < 3 \times 10^4$). For visualisation purposes, the rows and columns of the adjacency matrices in panels a and c of Fig. 3 have been arranged to highlight the block structure that results from modularity optimisation. In addition, rows and columns inside modules were sorted, in panels i and k, in order to highlight the possible nested structure within them[36,37]. Clear to the naked eye, each compartment presents an internal nested architecture[34]. This is a natural consequence of the node-level visibility-maximisation strategy as it adapts to system-wide environmental conditions: as long as these conditions are stable around weakly connected topics, nestedness emerges in those relatively isolated subsystems[38,40]. As soon as the boundaries across subsystems are blurred ($t > 3 \times 10^4$, panels b and d in Fig. 3), global nestedness prevails.

This subtle insight, which stems from the model, reframes the empirical findings presented above. Indeed, the information network is not swapping between two radically different architectures—often even antagonistic[47,52]—but rather fluctuating across nested self-similar arrangements at different scales. To quantify them, $\mathcal{N}$ is not a suitable tool, because it is designed to capture nestedness at the global scale only. For this reason, we resort to in-block nestedness $\mathcal{I}$[39,47,56], which generalises $\mathcal{N}$. On the one hand, when nestedness emerges at the global scale (one

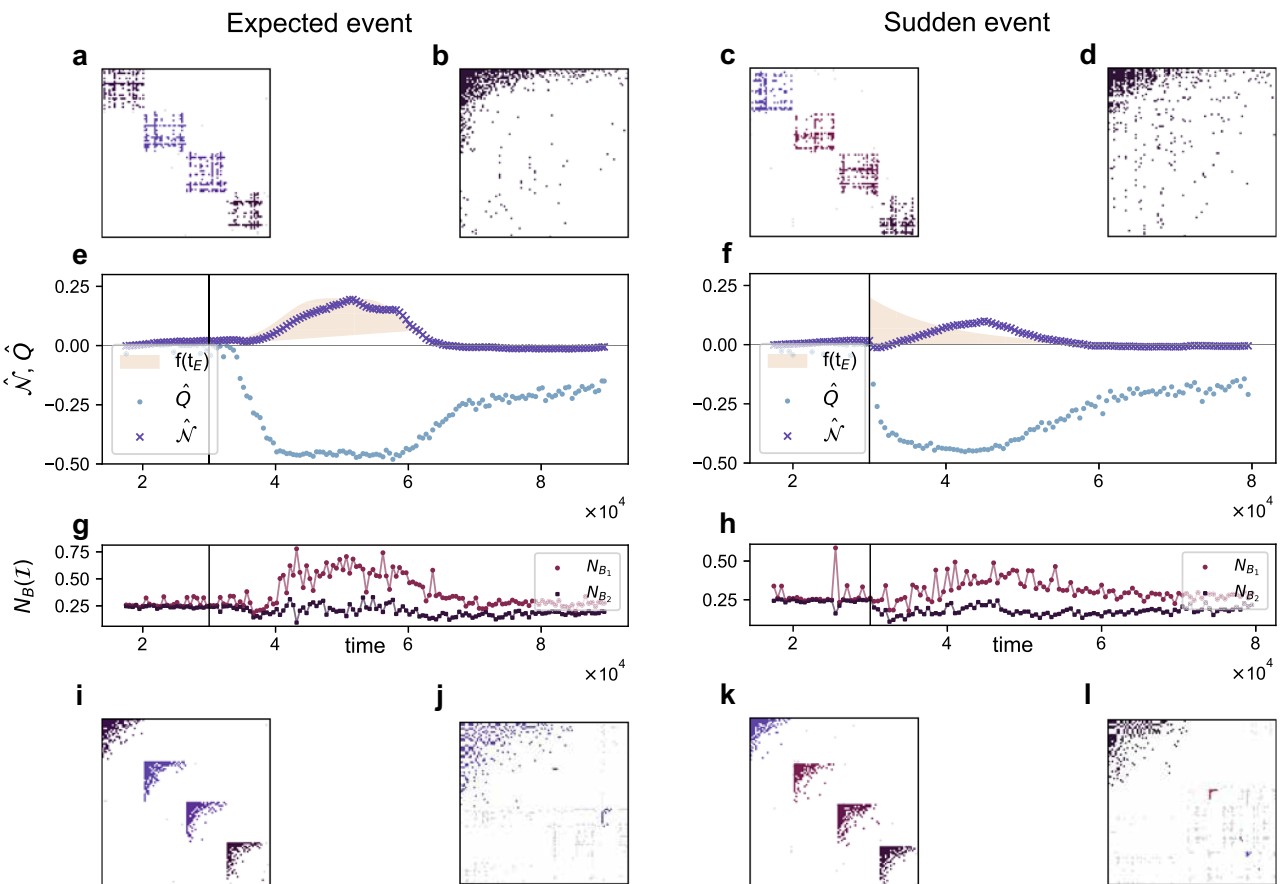

**Fig. 3 Structural evolution in the visibility optimisation model.** The figure corresponds to controlled numerical experiments at the stable stationary state, by holding fixed the number of species (100 users and 100 hashtags) and connectance (i.e., the fraction of non-zero interactions) $C \sim 10^{-2}$. The dynamics seek to maximise individual species abundances by varying the network architecture. Initially, links between users and hashtags are laid at random. Users and hashtags are aligned with a number of predefined topics ($T = 4$, as in Fig. 2). Panel **e** models the increase, sustainment and decay of attention in programmed events (e.g., election day), represented as a monotonically decreasing yellow shade starting at $t = 3 \times 10^{4}$. Panel **f**, instead, mimics the arrival of an unexpected exogenous event, represented also as a yellow shade. In the absence of an exogenous event, and following the trend observed in empirical data, the model initially organises in a clear block structure. Once the external event enters the system, the network blurs its modular organisation (smoothly in panel **e**; abruptly in panel **f**), and evolves towards a hierarchical, nested configuration. Such trend is visible also from panels **g** and **h**, where changes in $N_B(\mathcal{I})$ point at the emergence of a single large nested block. After the effects of the shock fade, the network slowly recovers its baseline modular configuration. The adjacency matrices surrounding the plots show the block and in-block nested structure of the bipartite network immediately before the onset of the perturbation (panels **a**, **c**, **i** and **k**, respectively), and the nested and in-block nested arrangement sometime after (panels **b**, **d**, **j** and **l**, respectively). In all panels, results correspond to an average of over 10 realisations.

block, $B = 1$), then we have that $\mathcal{I} = \mathcal{N}$. On the other hand, when the network presents several blocks ($B > 1$), each one arranged in a nested manner, then $\mathcal{I} > \mathcal{N}$.

It makes sense now to revisit the previous numerical and empirical results, now through the lens of in-block nestedness. Figure 1b and g, and Fig. 3g and h, respectively, monitor the relative size of the largest ($N_{B_1}/N$) and second-largest ($N_{B_2}/N$) nested blocks. In both empirical and numerical cases, we observe that nearly-perfect consensus is reached at different moments ($N_{B_1}/N \approx 1$), while a fragmented public sphere dominates most of the time. The relative size of the second-largest nested block ($N_{B_2}/N$) allows for an easier interpretation of the level of consensus reached at each time. Again, the analysis of additional datasets in Supplementary Note 1 confirms the generality of the results.

Our framework allows explaining the puzzling transition between partial and global consensus. A fast re-organisation from modular (nested) to nested (modular) architectures seems paradoxical and hard to achieve. Nevertheless, the system can swiftly adapt to any state of collective attention through an intermediary arrangement that combines the structural signature

of visibility maximisation with the existence of a fragmented public sphere.

**Lasting effects of perturbations on the system's dynamics.** Up to now, the focus on the macroscale and mesoscale has prevented us from connecting empirical observations with the model at the microscopic level. Here, we attempt to perform a comparison—even if qualitative—between the model and the data, exploiting the concept of abundance. The translation of the concept of abundance to the online communication context can be thought of as the number of times an item is present on screens. With language abuse, this is tantamount to the number of individuals (e.g., hashtag instances) that build up the species (e.g., *the* hashtag). Following this line of reasoning, for the empirical data on the hashtags' side, we can track the hashtag usage frequency over time, as a proxy for hashtag abundance from the model.

We compare the evolution of such abundance in the model and the data (Spanish and Nepal datasets). Panel a of Fig. 4 shows, from our numerical simulations, the changes in abundances of the hashtags over time, identifying with a colour the topic they are

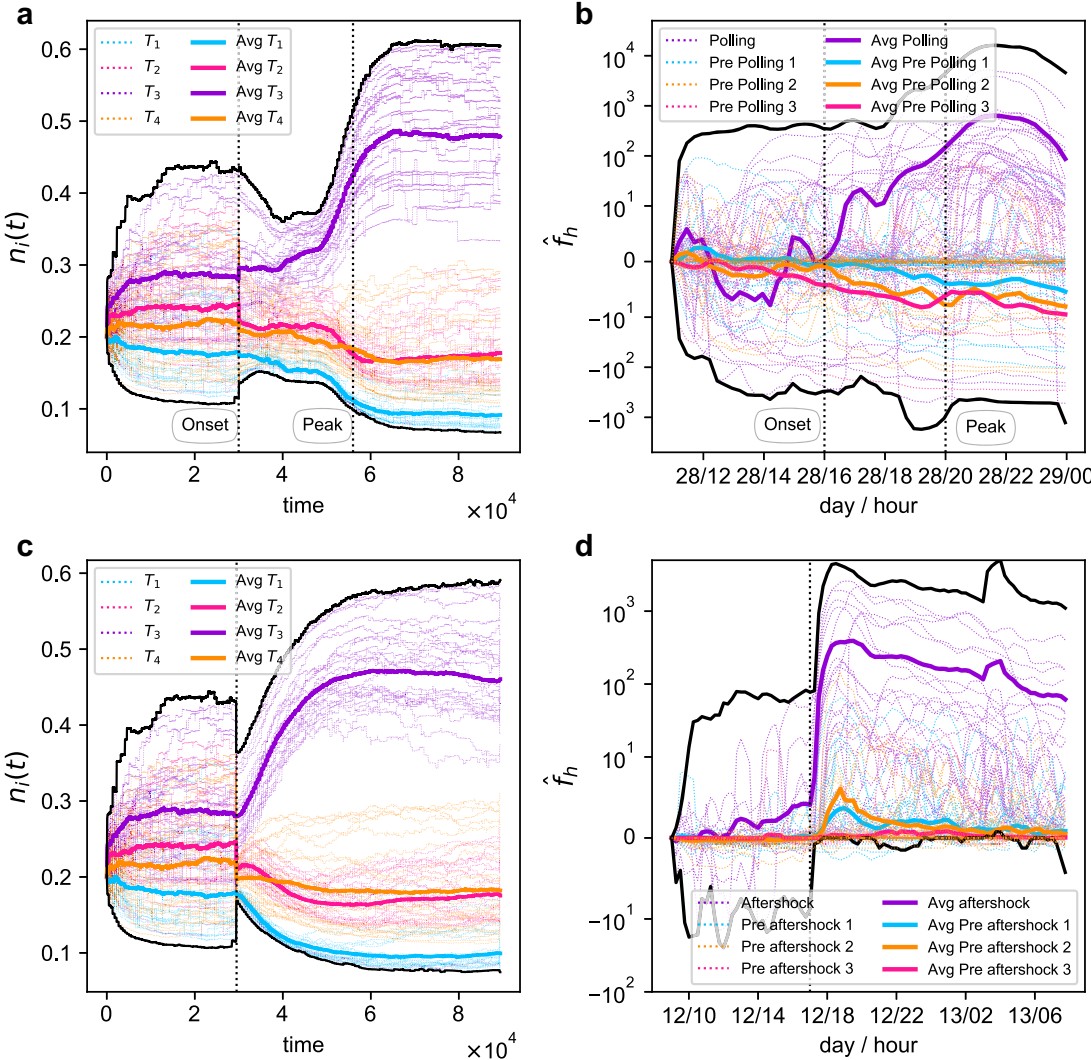

**Fig. 4 Evolution of abundances for a 4-topic information ecosystem.** Panels **a** and **b** correspond to synthetic and empirical expected events. For the numerical simulations, hashtags in $T_3$ (purple) begin a smooth increase in abundance $n_i$ at the event onset, which becomes steeper as the peak of the event approaches. Hashtags in other topics (blue, orange, fuchsia) experiment a slow decline. The same happens for the Spanish election day (panel **b**), regarding the evolution of hashtag frequencies $\hat{f}_h = f_h(t) - f_h(0)$, although admittedly with fluctuations. In this case, each colour corresponds to different communities, as detected from the networks maximising $Q$. Panels **c** and **d** correspond to synthetic and empirical unexpected events. Except for the abruptness in the increase of the purple hashtags in the Nepal dataset (much faster than its synthetic counterpart), the similarities are clear to the naked eye. Note that, in panels **b** and **d**, hashtag frequencies are shifted to their initial value. Remarkably, all four panels evidence that, at the microscale, a sufficiently strong perturbation impedes the system to recover the pre-event state, i.e., the system has achieved a new stable state. This result contrasts with the structural flexibility observed at the mesoscale and macroscale, in which the system remains within a narrow set of possible arrangements.

ascribed to. We observe that, prior to the event, the abundances of the hashtags are distributed rather uniformly within a narrow range. After the onset of the expected event, however, the abundance of the hashtags in topic $T_3$ (the one to which users' attention is shifted to) begins to increase. In the time range $3 \times 10^4 < t < 5.5 \times 10^4$ we observe a clear separation between the hashtags from $T_3$ with respect to the ones from the other topics. In the simulations mimicking expected events, the artificial shock peaks at $t = 5.5 \times 10^4$. Slightly before that time, hashtags in $T_3$ witness an even stronger increase up to $t = 6 \times 10^4$ (that is, beyond the peak time). After that, the system stabilises and appears to be unable to bounce back to the original, quite uniform abundances.

Panel b of Fig. 4 shows the usage frequency of actual hashtags over time in the Spanish dataset, where events are known in advance (in this case, election day on April 28). Adapting the logic of the model to empirical data, we show the trajectories of a group of

hashtags that belong to 4 different communities, the largest ones shortly before (light blue, orange, fuchsia) and at the time the ballots were closed (violet). The abundance of hashtags is represented here as their absolute frequency, shifted to the value at the beginning of the observation, i.e., $\hat{f}_h = f_h(t) - f_h(0)$, window to enable the comparison of their evolution. As in its model counterpart, the vertical lines show the buildup of conversations ahead of the results (around 4 p.m., "event onset" tag), and the electoral schools closing time (8 p.m., "event peak" tag). Overall, we observe a striking qualitative agreement between the simulated hashtags abundance (model) and the hashtag frequencies (data). Until 4 p.m., all 4 communities present a rather flat and uniform activity (note the logarithmic scale: apparently large fluctuations, e.g., between 12 p.m. and 2 p.m., imply frequency changes below 10). In the period 4 p.m.–6 p.m., the behaviour of the violet subset of hashtags resembles that of the hashtags of $T_3$ when the event occurs (slow but

steady separation from the other hashtags, with a frequency increase between $10^1$ and $10^2$); and also a more pronounced boost in the period 6 p.m.–10 p.m. (i.e., 2 hours before and after the event peak). The violet subset of hashtags clearly dominates the scenario even at midnight and starts an expected decline as conversations mostly halt during the late-night period. On the other hand, the subset of hashtags from the pre-debate stage (following $T_1$, $T_2$, $T_4$ in the model) present moderate decreases before 4 p.m., and losses are stronger after that time (especially light blue and orange topics).

For a complete picture, we study, as well an unexpected event. Panels c and d of Fig. 4 represent the evolution of abundances in an artificial setting with an unexpected event happening at $t = 3 \times 10^4$; and the evolution of hashtag frequencies around the time of Nepal's earthquake main aftershock (May 12, around 5 pm), respectively. Similar to its "expected" counterpart, our numerical experiments on panel c show a separation of the violet hashtags in $T_3$, with slight decreases of the other topics $T_1$, $T_2$ and $T_4$. The system also appears to be unable to return to the pre-event stage, and so the only obvious difference is that the separation occurs in an abrupt way. On panel d, we see the evolution of the frequencies of hashtags that belong to four of the largest communities detected in the data, slightly before (light blue, orange, fuchsia) and right after the aftershock (violet). Clearly, hashtags in the violet community present a sudden increase, followed by a very slow decrease resembling the one observed for $t > 6 \times 10^4$ in panel c. Given the international impact of the earthquake in Nepal, there is not decay during the night period.

These two examples extend the mesoscale and macroscale connections between data and model to the microscale. Furthermore, they provide a different perspective of our approach with regard to the memory of the system and the trace that exceptional events leave behind. From the mesoscale and macroscale, it is still valid to say that the system is trapped in a narrow set of structural configurations (namely, nested arrangements with only one or several blocks): this explains our use of the term "flexible". And yet, structural flexibility does not imply that the dynamic states of the system remain the same. Strong enough perturbations push the system away from its present stable state towards a new one. In this sense, the perturbation produces a long-lasting effect on the system's node dynamics.

## Discussion

The transit from low-bandwidth management of public information, to a decentralised and fragmentary scenario, has changed the way in which humans process information. In the context of a "cognitive bottleneck", the relevant drivers of online communication need to be identified. Building on diverse evidence, from cultural evolution[14–16] to neuroscience[5,7], we suggest that these drivers are competition for cognitive resources, mutualistic exploitation of the content, co-adaptation of actors' and memes' visibility, and environmental conditions—which heavily resemble the ingredients of natural mutualistic assemblages. So far, incursions in such an ecological mindset have been sparse[1,10,11,22]. In this work, beyond a metaphoric interpretation, we show that an ecological framework—with explicit use of competitive and mutualistic interactions as drivers of information dynamics—is a powerful tool to describe the evolution of information ecosystems. Indeed, although simple neutral models may account for emergent patterns in popularity[10,11] and attention distribution[57], we show that our non-neutral, niche-based population dynamics model can successfully explain the complex interplay when combining actor-meme systems, attentional niches and environmental shocks. We do so under the premise that the only condition that defines mutualism is the exchange of goods or services between two species, i.e., the fact that each species involved in mutualism must receive a benefit from the interaction. This is not to say that a one-to-one mapping is possible between natural mutualistic assemblages and online communication systems.

The success of the proposed approach is noteworthy at all the relevant scales of the system, i.e., macro, meso and micro. A fine quantitative fit is difficult to achieve at this stage since our synthetic approach is very simple in purpose: not only the presented toy model has an arbitrarily small size, but also a small amount of "topics", which, on top of that, are equally sized (exactly a fourth of the synthetic users and hashtags are centred around each topic). Clearly, all of this represents an idealisation of actual systems, and therefore some specific empirical particularities cannot be matched. Our modelling framework is a simple (but not too simple) way to understand the fundamental ingredients driving the observed emergent patterns in online communications systems.

Our results open an ambitious research alley. In the shorter term, future efforts should attempt to disentangle the apparent contradiction between structural flexibility, i.e., the propensity of the system's architecture to stay fixed at different in-block nested configurations, and dynamical instability, i.e., the fact that strong enough perturbations impede nodes to bounce back to their previous state variables. Also, further work should seek to mimic the microscopic dynamics of users' abundances before and after breaking events. These cannot be explained without including death-birth and invasion processes[58,59], which are in turn necessary to understand how influential users and viral contents emerge. Similarly, this initial proposal rules out "cultural drift"—the slower changes in the users' topical preferences—which leads to persistent structures and shapes communication flows.

Closer to practical aspects, there are transitions to/from nested/modular arrangements for which no explanation is provided (e.g., April 23 in Fig. 1e). It is often possible to find echoes of those transitions in the media, which explain their occurrence a posteriori. Whether those echoes can be found or not, the detection of structural changes is tantamount to the detection of collective attention foci. As such, our work contributes to the study of what is expected or unexpected, remarkable or unremarkable, in online communication streams—which remains an open question.

Reaching further, the tradition in theoretical ecology aimed at understanding and preventing the collapse of ecosystems can be adopted to decipher how social media and information bubbles shape our thinking[60], or, in the opposite direction to disrupt and break misinformation dynamics and polarisation. Related to this, we foresee as well a connection between the extensive research on stability and resilience in natural ecosystems and their informational counterparts. In this sense, we are convinced that such interchange of techniques and models could be beneficial for theoretical ecology too, as it will allow testing theories and methodologies in a more controlled, data-rich environment with a faster time scale at play.

## Methods

**Empirical and synthetic data**. For both synthetic and empirical cases, we represent a bipartite unweighted network as a $N_U \times N_H$ matrix $\mathbf{A}$, where rows and columns refer to users and hashtags, respectively. Elements, therefore, represent links in the bipartite network, i.e., if the element $a_{uh}$ has a value of 1, it represents that the user $u$ produced the hashtag $h$ at least once, otherwise $a_{uh}$ is set to 0.

For each empirical dataset, we split the Twitter stream into chunks according to non-overlapping time windows with three hours of duration $\omega = 3h$, containing the $N_U = 2000$ most active unique users, while the number of hashtags is variable (depending on the amount produced by those $N_U$ users). In this way, for each snapshot, a rectangular binary presence-absence matrix is created. During the onset of the events, the procedure was repeated considering time windows of 15 mins of duration. It is important to highlight that the networks may not contain the same nodes across $t$: as time advances, users join (disappear) as they start (cease) to show activity; the same applies for hashtags, which might or might not be in the focus of

attention of users. See Supplementary Note 1 for further details on the construction of the networks, including a discussion regarding the choice of $N_U = 2000$. Also, Supplementary Note 3 introduces a discussion on the temporal continuity of users and hashtags over time ('species turnover') in the data and the model.

For the generation of synthetic data, we set up a small network of 100 users and 100 hashtags, and the interactions between users and hashtags are laid at random with a connectance of $C \sim 10^{-2}$. The choice of such $C$ is meant to match the same order of magnitude of empirical networks' connectance when we take $N_U = 100$, following the logic of the work by Suweis et al.[23]; see Supplementary Note 2.

**Structural measures.** In this work, we explore the structural evolution of the networks by means of three arrangements, one at the macroscale (nestedness[45,46]), and two at the mesoscale (modularity[28], in-block nestedness[34,39,47]). We focus our attention on modular, nested and in-block nested patterns since all of them have been observed prominently in ecology[36,52,53,61] and in information systems[22,39]. In the following, we provide the definitions of the three measures.

*Nestedness.* The concept of nestedness appeared, in the context of complex networks, over a decade ago in systems ecology[31], and was previously introduced as a way to describe the patterns of distribution of species in isolated habitats[45]. In structural terms, a perfect nested pattern is observed when specialists (nodes with low connectivity) interact with properly nested subsets of those species interacting with generalists (nodes with high connectivity), see Supplementary Fig. 3a. Here, we quantify the amount of nestedness in information networks by employing an overlap metric[62] introduced by Solé-Ribalta et al.[39]:

$$\mathcal{N} = \frac{2}{N_r + N_c} \left\{ \sum_{i,j}^{N_r} \left[ \frac{O_{ij} - \langle O_{ij} \rangle}{k_j(N_r - 1)} \Theta(k_i - k_j) \right] + \sum_{l,m}^{N_c} \left[ \frac{O_{lm} - \langle O_{lm} \rangle}{k_m(N_c - 1)} \Theta(k_l - k_m) \right] \right\},$$

(1)

where $N_r$ and $N_c$ correspond to the number of rows and column nodes, respectively. The values $O_{ij}$ (or $O_{lm}$) measure the degree of links overlap between rows (or columns) node pairs; $k_i, k_j$ corresponds to the degree of the nodes $i,j$, and $\Theta(\cdot)$ is a Heaviside step function that guarantees that we only compute the overlap between pair of nodes when $k_i \geq k_j$. Finally, $\langle O_{ij} \rangle$ represents the expected number of links between row nodes $i$ and $j$ in the null model and is equal to $\langle O_{ij} \rangle = \frac{k_i k_j}{N_r}$.

*Modularity.* The modular structure is a rather ubiquitous mesoscale architecture[29,63–65] and implies that nodes are organised forming groups, i.e., devoting many links to nodes in the same group, and fewer links towards nodes outside[66], see Supplementary Fig. 3b. Given the huge number of possible ways to partition a graph into groups, an exhaustive assessment of every partition's fitness is unfeasible. Hence, scholars have developed several algorithms that are able to find fairly good approximations or (sub)optimal partitions, by means of the optimisation of a fitness function[28,67–69]. Here, we search for a (sub)optimal modular partition of the nodes by applying the extremal optimisation algorithm[67] to maximise Barber's[70] modularity, which is an extension of the original formulation introduced by Newman[28], to bipartite networks:

$$Q = \frac{1}{L} \sum_{i=1}^{N_r} \sum_{j=1}^{N_c} \left( \tilde{A}_{ij} - \tilde{p}_{ij} \right) \delta(\alpha_i^r, \alpha_j^c)$$

(2)

where $L$ is the number of interactions (links) in the network, $\tilde{A}_{ij}$ is the adjacency matrix which denotes the existence of a link between rows and columns nodes $i$ and $j$, $\tilde{p}_{ij} = k_i k_j / L$ is the probability that a link exists by chance, and $\delta(\alpha_i, \alpha_j)$ is the Kronecker delta function, which takes the value 1 if nodes $i$ and $j$ are in the same community, and 0 otherwise.

*In-block nestedness.* Nestedness and modularity are emergent properties in many systems, but it is rare to find them in the same system. This apparent incompatibility has been noticed and studied, and it can be explained by different dynamical pressures: certain mechanisms favour the emergence of blocks, while others favour the emergence of nested patterns. Following this logic, if two such mechanisms are concurrent, then hybrid (nested-modular) arrangements may appear. Hence, the third architectural organisation that we consider in our work refers to a mesoscale hybrid pattern, in which the network presents a modular structure, but the interactions within each module are nested, i.e., an in-block nested structure, see Supplementary Fig. 3c. This type of hybrid or "compound" architectures was first described in Lewinsohn et al.[34] and has been further explored in the last decade[35,37–39]. Using the formulation developed in ref. [39], the degree of in-block nestedness of a network $\mathcal{I}$ can be computed as

$$\mathcal{I} = \frac{2}{N_r + N_c} \left\{ \sum_{i,j}^{N_r} \left[ \frac{O_{ij} - \langle O_{ij} \rangle}{k_j(S_i - 1)} \Theta(k_i - k_j) \delta(\alpha_i, \alpha_j) \right] + \sum_{l,m}^{N_c} \left[ \frac{O_{lm} - \langle O_{lm} \rangle}{k_m(S_l - 1)} \Theta(k_l - k_m) \delta(\alpha_l, \alpha_m) \right] \right\},$$

(3)

where $S_i$ accounts for the number of nodes in the same guilds of $i$ and that belong to the same community. Worth highlighting this hybrid structure reframes nestedness, originally a macroscale feature, to the mesoscopic level. In this sense, by definition, $\mathcal{I}$ reduces to $\mathcal{N}$ when the number of blocks is 1.

For the community analysis, we apply a variant of the extremal optimisation algorithm[67] adapted to maximise both, Barber's bipartite modularity, and the in-block nestedness function[47]. Supplementary note 4 discusses the convenience and quality of this maximisation heuristic.

**Niche model and population dynamics.** The model is developed for a bipartite network representing users and hashtags as two interacting guilds (denoted $U$ and $H$). Each species $i$ has to assign a niche profile, formulated as a Gaussian function $G_i(s)$ with width $\sigma_i$, positioned according to a number of $T$ topics of their interest, that is created equidistant on the niche axis, see Fig. 2. To perform the numerical simulations, we employ a model that follows a Lotka-Volterra dynamics, with Holling-Type II mutualistic functional response[23,54]:

$$\frac{dn_i^U}{dt} = n_i^U \left( \rho_i^U - \sum_j \beta_{ij}^U n_j^U + \frac{\sum_k \gamma_{ik}^{UH} n_k^H}{1 + h\sum_k \theta_{ik}^{UH} n_k^H} \right),$$

$$\frac{dn_i^H}{dt} = n_i^H \left( \rho_i^H - \sum_j \beta_{ij}^H n_j^H + \frac{\sum_k \gamma_{ik}^{HU} n_k^U}{1 + h\sum_k \theta_{ik}^{HU} n_k^U} \right).$$

(4)

Here, the coupling matrices $\boldsymbol{\beta}$ and $\boldsymbol{\gamma}$ define the competitive (within guild) and mutualistic (across guild) interactions, respectively. In this context, $\beta$ encodes the competition for others' attention (among users), and for the cognitive resources of speakers (among hashtags). Both interaction matrices depend linearly on the niche overlap between pairs of species $G_{ij}^{gg'} = \int G_i^g(s) G_j^{g'}(s) ds$. In addition, these matrices have a global factor, $\Omega_m$ or $\Omega_c$, which tune the strength of mutualistic or competitive interactions, such that, $\gamma_{ik}^{UH} \propto \Omega_m \cdot G_{ik}^{UH} \cdot \theta_{ik}^{UH}$ and $\beta_{ij}^{HH} \propto \Omega_c \cdot G_{ij}^{HH} \cdot \lambda$, respectively. The competitive matrices include an additional parameter $\lambda$, that helps to balance the inter–intra topic competitive interactions: see Supplementary Note 2. Finally, $\boldsymbol{\theta}$ is the adjacency matrix, and $h$ is the handling time of the Holling-Type II mutualistic functional response. Within the information ecosystem context, these equations represent a phenomenological way to describe the evolution of the nodes' visibility as a function of their interaction. In particular, $n_i^U$ may represent the number of instances in which user $i$ is present in other users' screens, while $n_j^H$ may quantify the popularity of a given hashtag $j$. Assuming that preferential attachment mechanisms of various types affect the nodes visibility, $\rho_i^U$ and $\rho_i^H$ model the associated exponential growth (if they are positive). The handling time $h$ effectively models the constraint that users cannot interact with a very large number of hashtags due both to time and character constraints. Due to these limitations, the benefit obtained through mutualistic interactions does not grow monotonically with the number of partners.

**Optimisation proccess.** On the dynamical side, we also introduce an optimisation principle, in which users attempt to change their mutualistic partners (memes) in order to maximise their visibility. Specifically, we start a rewiring adaptation process following the approaches in Suweis et al.[23] and Cai et al.[25]:

*Rewiring.* at constant time intervals, a random species $U$, with a least one link, is selected and rewired to a randomly selected species $H'$, removing one of its previous links $H$, with probability $p_{UH} \propto 1 - k_H^{-1}$, in order to maximise their individual abundances.

*Link recovery.* At the end of each time interval, if the current abundance is greater than the previous one, the current (new) link is kept; otherwise, the previous link is recovered.

Note that in this scheme, memes do not optimise any quantity. Thus, the rearrangement of the memes network structure is simply a byproduct of the user's actions.

**Introduction of external events.** Finally, our dynamic model includes as well a mechanism to introduce exogenous events in the environment. We modelled these events as transitory shifts in the users' attentional niches. Concretely, we changed the user's niche centres towards a single common topic for a limited period of time. After some time, we move them back to their original niche topics. In the simulations, the users' niches were modified in the following way:

$$G_i^E(s) = [1 - f(t_E)] G_i(s) + f(t_E) G^{E'}(s),$$

(5)

where a user's niche becomes a composition of two Gaussian niches: $G^{E'}(s)$ corresponding to the general event, $G_i(s)$ that is the original one corresponding to the user's intrinsic interests, and $f(t_E)$ is the function that governs the growth and decay of the external event. In this work, we have modelled two profiles along the lines of Lehmann et al.[42], see Supplementary Note 2 and Supplementary Fig. 8.

Simulations were performed by integrating the system of ordinary differential equations using a fourth-order Runge–Kutta method. We assigned the same initial abundance $n_0 = 0.2$ and intrinsic growth rates $\rho_U = \rho_H = 1$ to all users and hashtags. Species were considered to suffer extinctions when their abundance density was lower than $10^{-4}$. Finally, the handling time $h$, of the Holling-Type II mutualistic functional response was set to 0.1. See Supplementary Note 2 for further details.

**Reporting summary**. Further information on research design is available in the Nature Research Reporting Summary linked to this article.

## Data availability

The Catalan and Spanish datasets are available at OSF with the identifier https://doi.org/10.17605/OSF.IO/J5QWX. The rest of the datasets employed in this study were collected by Zubiaga[51] and are available at figshare with the identifier https://doi.org/10.6084/m9.figshare.5100460.v2.

## Code availability

The software code for nestedness measurement, and modularity and in-block nestedness optimisation, is available at https://github.com/COSIN3-UOC/N-Q-IBN with https://doi.org/10.5281/zenodo.4557009. The code for simulations of the dynamical model is available at https://github.com/COSIN3-UOC/dynamical-niche-model with https://doi.org/10.5281/zenodo.4555890.

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

## Acknowledgements

M.J.P., A.S.-R. and J.B-H. acknowledge the support of the Spanish MICINN project PGC2018-096999-A-I00. M.J.P. acknowledges as well the support of a doctoral grant from the Universitat Oberta de Catalunya (UOC). S.S. thanks the support of UNIPD through ReACT Stars 2018 grant. S.M. and V.C.-S. acknowledge partial financial support from the Agencia Estatal de Investigacion (AEI, Spain) and Fondo Europeo de Desarrollo Regional under Project PACSS Project No. RTI2018-093732-B-C22 (MCIU, AEI/FEDER, UE) and through the María de Maeztu Program for units of Excellence in R&D (MDM-2017-0711). C.A.P. acknowledges the support provided by grant No. ANR-18-CE30-0013 from the Agence Nationale de la Recherche. All authors acknowledge the support to the TEAMS project of the Cariparo Visiting Program 2018 (Padova, Italy). All authors thank Joan T. Matamalas from the Universitat Rovira i Virgili for the help with the acquisition of the Spanish elections dataset.

## Author contributions

All authors designed research. M.J.P., A.S-R. and J.B.-H. collected and curated the data, and performed research. All authors analysed the results. J.B.-H., S.S and S.M. wrote the paper. All authors approved the final version.

## Competing interests

The authors declare no competing interests.
