## [Peer Review File · Nature Communications]

Reviewers' Comments:

Reviewer #1:

Remarks to the Author:

In their manuscript "Resilience and elasticity of co-evolving information ecosystems" the authors report an empirical observation of the structural reactions of bipartite User-Hashtag networks to external events, specifically a switch from modular structures to nested structures and later refine that picture as a switch from local nestedness to global nestedness. For the modelling approach, they use Lotka Volterra dynamics, with competition among Users/Hashtags and mutualistic coupling between Users and Hashtags, which they dynamically alter to account for external events. On top of these dynamics they model an optimization procedure, where rewiring in the network happens to maximize abundance.

While it is interesting to investigate how information ecosystems react to external events and the modelling framework is useful, there are shortcomings in the empirical analysis and unclear relations to the model and comparisons to alternative hypotheses, which need to be addressed, in order to allow assessing the innovative and explanatory value of the present manuscript. I am listing my main remarks in the following.

Main remarks:

Contribution of the manuscript: In the introduction, the connections to existing literature needs to be made more explicitly. Currently, it is not clear what parts of the presented work are new and which built on existing work about effects of external events on the information ecosystem and modelling of the attention economy (e.g., refs. 6, 7, 12, 14, 39, was the shift of modularity and nestedness also reported in ref. 12?). Explicitly relating them to the present work and stating the new contribution of the present manuscript explicitly is needed.

Empirical data: The representation of the data, as bipartite networks between users and hashtags is useful, but needs more systematic/statistical analysis beyond the qualitative reporting of the observation of changing modularity vs. nestedness.

The first and most striking observation, the increased activity needs to be discussed more. How does this affect the bipartite networks, i.e., how is the increased activity distributed among users and hashtags? (histograms, network plots?) While the two measures are interesting, the networks and their change can be characterized more detailed. Also, how does, for example, the composition of hashtags change during the events?

Is it true that nestedness and modularity are exclusive in all regimes, how do they relate to network density? Especially, because the Null model here could be that the activity (however it is distributed, e.g., just randomly) is just quantitatively increasing, which could be driving the observations e.g., of reduced modularity. This needs to be tested and the measures used need to be compared to other observables to allow arguing for the modelling approach.

Also, further observables would allow more quantitative comparisons with the model (e.g., degree distributions etc.), which stay currently qualitatively.

Modelling framework: Here, the connections to the observations need to be made much more explicit. Which quantities are conserved in the model, which in the data (e.g., are the number of connections preserved in the model? Are they in the data?)? Referring to the remark above, as the change in the activity seems to be the dominant observation, it needs to be explicitly addressed in the model.

Similarly, the number of hashtags is constant in the model, while it changes in the data. How does it change in the data? Can this be incorporated in the model? Again, quantitative comparisons would help to show the connection of the model with the data, which has a sufficient complexity to be more explicit here.

While the "abundance" stays a theoretical concept without counterpart in the data (likes would be a measure?), other parts of the model can be explicitly compared (e.g., distributions of links,

activity distribution, which is currently uniform in the model?).

Relatedly, when the external shocks are incorporated the distinction between expected and sudden events is made, but are they distinguished in the empirical data. In Fig. 1 the two types are shown but the reaction of the system does seem to be most evident in the activity, rather than the other observables. This is used to model the attentional shift function in the supplementary material (the most explicit relation to the data), but compares different observables, right? But an increase inactivity does not necessarily reflect a shift of attention. Referring to an earlier point, to test the null model of randomly increasing activity, this can be checked in the data to back up this assumption.

The in-block nestedness is an interesting measure, but seems to be quite sensitive to the community detection method, that should be addressed.

Minor remarks:

- The in-block nestedness needs to be introduced earlier
- A table or list, introducing/listing the variables/ingredients of the model and their counterparts in the data would help a lot
- The caption of Figure 2 is not self-explanatory
- In the main manuscript (without the supplementary material) it is difficult to follow the structure of the model, better to pull some of it in the main text or in the methods section.
- How long were the snapshots to create the empirical networks?
- Typo in page 7 "...adjacency matrix. , and h is the..."

Reviewer #2:

Remarks to the Author:

The authors study the structure and dynamics of bipartite network of hashtags and twitter users. I found the approach fascinating and apologize for taking a bit long to produce my review. It simply took me quite some time to get a feel for what is really the result, and how it is obtained. My opinion should not be weighted too heavily as I am not a network scientist. So, please take my views as observations that may hopefully be of some guidance for revising this for the current or another journal.

The title promises an insight in resilience and elasticity, and later the authors mention 'remarkable structural elasticity'. However, it does not seem surprising to me that people start talking immediately about a big event when it happened (elasticity), and then return to their normal interests (resilience). In nature, bipartite networks of species involve adaptations such as evolutionary tuning between pollinator and specific flower shapes. That takes time, and is quite an investment. Using a hashtag and then dropping it is obviously swifter. In short, I do not see an eye-opener when it comes to the terms in the title.

Also, I am not surprised by the nestedness vs modularity results. Isn't it a bit obvious that people from different groups and interests start talking about the same big thing when it happened? This would be visible in all kinds of indicators including nestedness, but also other indicators such as abundance of the most mentioned hashtag, Shannon diversity of hashtags, or network connectivity measures? Thus, what is really the surprise here?

This may sound a bit negative, but I do think the information the authors analyze is fascinating. I would just think that other research questions could perhaps be more promising.

If big events do get everyone talking about the same thing, does this leave a trace in social cohesion? Are groups becoming connected in a meaningful lasting way? To me that would be a deep impact. So the fact that you do not seem to see that could imply that the nestedness and enhanced connectivity is not really meaningful? So big events do not really connect?

Or are some kinds of events more connecting than others? Could you detect that?

Reviewer #3:

Remarks to the Author:

I am surprised to see several new concepts frequently used in this paper including its title, but the authors often choose these terms without specific and valid explanations. While the term "resilience" has often been used across scientific fields, the term "elasticity" is not the concept with which (social) scientists are familiar. As a matter of fact, the first paragraph starts with several undefined terms and/or phrases including bio-cognitive limits, quotidian communication processes, oligopolistic media environment, attentional resources of the audience, etc. I am not criticizing the authors' language skills in English, but pointing out what core concepts are and what the surrounding terms are. Relations among conceptual words should have been revealed in a clearer way. Instead of "resilience" used in the title, I suggest that the authors choose "modular" VS "nested" architecture as the core concept of this paper. Thus, introduction should begin with them at the center of the writing. Public attention toward elections increase over time, but when an earthquake occurs, social interest explodes and disappears immediately. The choice of two datasets will significantly affect the findings. Last but not least, Han Woo Park's team has conducted numerous social media research in election and/or environmental disaster contexts including a recent Covid-19. You might like to take a look at and refer to their research.

Reviewer #4:

Remarks to the Author:

First of all I would like to apologize for my long delay in sending my review. This manuscript fits to the long-lasting quest searching for general patterns in complex systems. Thus, the questions addressed are timing and the patterns reported and the approach used very interesting. Having said that, as an ecologist and evolutionary biologist, my main concern is on the comparisons with ecosystems the authors used. In short, I think the ecological analogies used at the best inaccurately described and maybe flawed, imperiling the implications of the study to ecology and to the synthesis between theoretical ecology and applications in other scientific fields.

Main comments

1. The underlying reasoning of the manuscript is based on the idea of actor-meme coevolution. However, as far as I understood, memes are not shifting their patterns of interaction to maximize any quantity. They are passive resources that users interact. So, coevolution - in a biological sense, see my comment #2 - does not seem to me a valid benchmark. At the best, it is analogous to evolutionary resources against fixed resources and how it is modulated by other environmental factors.
2. Introduction: "Under the light of these four drivers –competition, mutualism, co-evolution, environment–, online communication systems become a special case of mutualistic ecosystems". This is an inaccurate statement. First, the citation here is misleading since the paper cited does not address online communication systems but the network structure of mutualistic assemblages. Second, there is no such thing as "mutualistic ecosystems". Mutualistic interactions are ecological interactions in which individuals of different species benefit from the interaction. Mutualistic interactions may lead to evolutionary or demographic effects at the population level. In some types of interaction, mutualisms may involve individuals from multiple species, forming mutualistic assemblages. Ecosystems, in turn, are systems in which organisms and environment are connected by pathways in which matter and energy flow. I have to say that this lack of accuracy with biological terms is not expected in a study that aims to connect different fields and similar misuse of ecological terms led, in the past, to multiple research dead-ends and simply wrong ideas that create more confusion than light (e.g., Gaia hypothesis). Finally, there are multiple differences between online communication systems and mutualistic assemblages related to spatial, temporal, and organizational scales. In the best-case scenario these two systems share similarities

that may drive the dynamics of a given state variable but I do not see how online communication systems are a special case of mutualistic assemblages.

3. The use of the term coevolution. Coevolution here is used, if I got correctly, in the sense of the feedback between the dynamics of networks (structural change across time) and the dynamics on networks (flow of information), i.e., adaptive or coevolutionary networks sensu Gross and collaborators. Coevolution, in evolutionary biology, however, is the reciprocal change of traits driven by natural selection in interacting populations of different species. In this context, coevolution can occur – and indeed is usually reported – in pairwise interactions. Again, the lack of accuracy when using the biological terms may imperil the analogies drawn between communication and biological systems. For instance, coevolution – in its biological sense - is just one of the processes that lead to this feedback between the dynamics of networks and the dynamics on networks in ecological assemblages. Non-evolutionary, behavioral changes are well known to lead changes in ecological interactions, as exemplified by the theoretical work by Peter Abrams.

4. Results: the notion that nestedness and modularity can coexist in modules with internal nestedness is not new. More than 10 years ago, Thomas Lewinsohn and his coauthors proposed these combined network structures. There are multiple empirical and theoretical studies exploring these patterns in ecological systems.

Minor points

1. In addition to inaccurate use of biological concepts, the authors imperil accuracy and precision at multiple parts of manuscript. A non-exhaustive list of examples include but are not limited to:
 - a. Abstract: “apparently chaotic and noisy”
 - b. Introduction: entire paragraphs without any reference supporting statements (see minor comment #3).
 - c. Results: “Biased as it may be. Twitter is without a doubt a sensitive platform that mirrors, practically without delay, exogenous events occurring in offline environments”. “vision of the non-virtual world”, “highly fluctuating nature of this endless communication stream”.
2. Abstract: “selective pressure, i.e. the chances to persist and reach widely are tightly subject to changes in the communication environment” → although selection may change across environment, selection may act even without environmental change.
3. Page 1: second and third paragraphs are essential to the paper but no references support the claims made.

Nature Communications - response to reviewers

Manuscript Number: NCOMMS-20-18890-T

Resilience and elasticity of co-evolving information ecosystems

We thank the four anonymous referees for their insightful and challenging comments, which we try to address below point by point. Whenever our responses imply changes in the main text or the supplementary information, we highlight those changes with blue coloured text.

Reviewer #1:

In their manuscript "Resilience and elasticity of co-evolving information ecosystems" the authors report an empirical observation of the structural reactions of bipartite User-Hashtag networks to external events, specifically a switch from modular structures to nested structures and later refine that picture as a switch from local nestedness to global nestedness. For the modelling approach, they use Lotka Volterra dynamics, with competition among Users/Hashtags and mutualistic coupling between Users and Hashtags, which they dynamically alter to account for external events. On top of these dynamics they model an optimization procedure, where rewiring in the network happens to maximize abundance.

While it is interesting to investigate how information ecosystems react to external events and the modelling framework is useful, there are shortcomings in the empirical analysis and unclear relations to the model and comparisons to alternative hypotheses, which need to be addressed, in order to allow assessing the innovative and explanatory value of the present manuscript. I am listing my main remarks in the following.

Main remarks: (we have numbered the comments for easier traceability)

(1) Contribution of the manuscript: In the introduction, the connections to existing literature needs to be made more explicitly. Currently, it is not clear what parts of the presented work are new and which built on existing work about effects of external events on the information ecosystem and modelling of the attention economy (e.g., refs. 6, 7, 12, 14, 39, was the shift of modularity and nestedness also reported in ref. 12?). Explicitly relating them to the present work and stating the new contribution of the present manuscript explicitly is needed.

We thank the reviewers for this comment, allowing us to better highlight the original and novel contribution of our work.

Indeed, we agree that there are, in the literature, some key works that mark a path towards an "info-ecological" understanding of online communication dynamics, and these contributions need to be distinguished from the current work. Our main point here is that those references only grasp partial aspects of the picture.

Refs. [6,7] (the latter represents a formalization and analytical development of the former), [9,10] in the new version, pay attention only to the competitive aspect of the problem (among users with limited attention), while the relationship to (and among) memes is missing. Ref. [14] (ref. [18] in the new version), complementary, focuses on the persistence of memes, and explicitly takes into account the (increasing) competition among them --but discards the users' side. All in all, these unipartite accounts of online communication miss the fundamental role of the user-meme interaction.

Ref. [12] ([15] in the new version), admittedly, is the most overlapping to the present work inasmuch it focuses on nestedness and modularity. And yet, besides this, it misses the most central contributions of the current work. Specifically:

- (1) Ref. [12] is founded on an observational analysis, and the result of that data analysis renders a qualitative discussion that hints to the plausibility of applying mutualistic population dynamics to information ecosystems. Nonetheless, the authors were not able to explain, with a comprehensive modelling approach, why the system shifts from one structure to the next.
- (2) Ref. [12] analysed in depth only one dataset (with a more shallow one on a general Twitter stream, i.e. not focusing on any topic in particular).
- (3) Ref. [12] did not introduce attentional and semantic niches, which play a major role in driving competitive and mutualistic relationships.
- (4) Ref. [12] did not identify, nor quantify, the presence of in-block nestedness in the system, which appears to facilitate the rapid shifts between two apparently distant structural organizations, such as nestedness and modularity.
- (5) Also, ref. [12] could not analyse the complete cycle of elastic behaviour of the system at the mesoscale, i.e. the tendency of the network to recover a modular architecture in the absence of system-wide external stimulus. The reason for this was a practical one, due to the lack of data at the time (that is, a dataset covering the whole range of the studied event).

Finally, ref. [39] (ref. [36] in the new version) is mostly unrelated to our research, and we cite it as a useful classification of events on Twitter. If anything, our results, both empirical and theoretical, suggest that the difference between those classes is subtle (affecting the abruptness of a transition), at least in relation to the structural stages and changes that the network undergoes.

Summarizing, the current work integrates partial observations in the past literature, and tries to encapsulate those scattered contributions into a unifying framework, uncovering new findings from it. We have tried to clarify and highlight the value and novelty of this work, introducing some changes in the introduction.

(2) 2.1. *Empirical data: The representation of the data, as bipartite networks between users and hashtags is useful, but needs more systematic/statistical analysis beyond the qualitative reporting of the observation of changing modularity vs. nestedness.*

We agree with the referee that other analyses, besides the strictly structural ones, are needed. Following this idea, we have worked on many additional insights. Some of them are covered here, but also in other related concerns raised by the referee later on (e.g. comments 3-7). Also, one of the main general concerns of the referee is related to the weak connection between the data and the model throughout the work. To strengthen such connection, most of the new findings for the data come along with the same analysis for the model (synthetic) counterpart.

Statistical insights

We thank the referee for noticing that, indeed, we had left the observation of changes in Q and \mathcal{N} at the observational level. At most, in the main text it is said that both quantities are “anticorrelated”, without any quantitative justification. We wrongfully overlooked this aspect. Therefore, we now validate that the relationship between both measures is indeed anticorrelated, both for the empirical data and the model’s outcome. Table R1 below reports the Pearson correlation between Q and \mathcal{N} across time. For the synthetic cases (last two rows), it is measured from $t = 1.5 \times 10^4$ onwards, to avoid the initial random fluctuations. For an even finer comparison, correlation is measured over the whole period covered for the datasets, but also before/during/after the main events that we identify in the manuscript (Fig. 1 of the main text, Fig. S4 of the SI).

Dataset	Whole period	Pre event	Event	Post event
Spanish elections	-0.8264	-0.2914	-0.9094 debate -0.7178 polling	-0.8467
Nepal Earthquake	-0.7337	-0.26566	-0.9358	-0.74138
Catalonia	-0.1490	-0.1036	-0.21326 diada -0.3079 polling	-0.7234
Euro	-0.7930	-0.42895	-0.8675 semis -0.7545 finals	-0.8827
Charlie Hebdo	-0.9180	-0.4496	-0.8979	-0.8240
Hong Kong	-0.5774	0.0034	-0.6194 occupy	-0.6180
Model “expected event”	-0.743	0.1625	-0.7611	-0.8641
Model “sudden event”	-0.6651	-0.0558	-0.7390	-0.8533

Table R1: Pearson coefficients for Q and \mathcal{N} at different times, for both, the model and the data.

Except for a single dataset (Catalonia), the matching between empirical (upper rows) and synthetic results (last two rows) is remarkable, not only in the periods where the correlation is strong, but also during the pre-event stages (where, in most cases, both correlations are irrelevant, despite their opposed signs).

Catalonia's mismatch with respect to the model, and to the other datasets, is obvious *during* the event (affecting as well the measurement for the whole period). We note that this dataset contains a number of differences with respect to the others:

1- It is by far the smallest dataset in the collection, and as such the most exposed to be affected by noise (see points 4 and 5 below).

2- It deals with a mostly local political event, which at the time (2014) had not yet attracted international attention. In this sense, the collection includes a mixture of tweets in Catalan and Spanish, and relatively few from outside Spain.

3- The Catalan political conflict is an enduring problem in Spanish politics, and has an associated extremely high polarization. Unlike the other datasets, which are mostly characterized by consensus (e.g. solidarity for those affected by the Nepal earthquake, Hong Kong protests or Charlie Hebdo massacre), the Catalan one is characterized by conflict: the dataset contains both users in favour and against a referendum, and a large fraction of Spanish users who think that a referendum should not even be discussed at all.

4- As a consequence of the previous point, the polling day contains not only tweets paying attention to the results of the (illegalised) referendum, but also many messages calling for political and legal action against the organisers, or simply appealing to a political dialogue between the parts. In structural terms, this translates into an in-block nested structure suggesting a sort of "partial consensus" among the different sides that participate in the conversation. In this situation, we can obtain intermediate values of both quantities (Q and \mathcal{N}), which implies a weaker anti-correlated behaviour.

All these explain the weak anticorrelation observed that day ($r = -0.3079$).

We have inserted this table, alongside explanations, in the Supplementary Materials.

Statistical significance of Q and \mathcal{N}

To strengthen the validity of our results further, we start by trying to refute a possible lack of statistical significance of Q and \mathcal{N} (which has been and is a controversial issue for these descriptors). The first thing to keep in mind is that the two descriptors incorporate a null model term in their definition. While this is not new for Q (its quantification has always been in reference to a suitable null term), it is also true for nestedness as we define it (which differs from mainstream definitions that do not include a random expectation term; see eq. (1) in the main text). Still, for the sake of completeness, we show below (Figure R1) the results of the two descriptors (\mathcal{N} and Q) not in their absolute values, but as z-scores against an ensemble of randomisations of the networks at stake (only for the Spanish dataset, due to computational limitations):

Figure R1: z-scores for modularity and nestedness, against an ensemble of 150 randomisations. Note that modularity falls below significance only during exceptional events both in the empirical case (left) and the model (right). This is not the case of nestedness, which remains always above the significance level (and yet, it experiments surges in z-scores during important events).

For each one of the matrices of the Spanish dataset we have generated 150 randomizations in which we preserve the link density. This form of randomization amounts to considering that the overall system's activity is kept, but users lack a preference for one or another meme for communication purposes. The black solid lines in the plots correspond to $z\text{-score} = 2$. The dotted lines show the actual \mathcal{N} and \mathcal{Q} values on the real matrices, that are indicated by the secondary y-axis in both panels.

In the left panels, we observe that during the debate the measured values for \mathcal{Q} are no longer statistically significant at the selected confidence interval, which is an expected result. For the second event (polling), we observe an abrupt decrease in the z-values for the measured \mathcal{Q} , although it is still significant under the considered threshold. In the case of nestedness, we observe that its values remain statistically significant for the whole dataset. Nonetheless, we find evident variances between their statistical significance during the crucial periods of the datasets. In general, the z-scores for \mathcal{N} are extremely high during the extreme event periods, despite low values (compared to the peak) outside the exogenous events. Pursuing stronger connections between findings in the data and the model, we have also analyzed, under the same scope, the outcome of our model. Results in the right panels of Fig. R1 show the same observed behaviour: during the artificially introduced event, the z-scores for \mathcal{Q} fall below statistical significance; while \mathcal{N} is significant overall the period, but with a marked surge after the exogenous introduction of an event at $t = 3 \times 10^4$.

Yet another way of approaching the significance of the structural transitions is to compare the results to the median in the time series. In the left column of Figure R2 we represent, for the Spanish dataset, the evolution of \mathcal{Q} (top) and \mathcal{N} (bottom) in time. The right column corresponds, conversely, to the results obtained from the model. We have measured, for all these sequences, the median and interquartile range (IQR), solid line and shade, respectively.

In the case of Q , in the empirical case (left), it only falls below $Q - IQR$ during the first (debate) and second (election day) large events, implying a large deviation from its baseline. The same can be said considering the results of our model: Q only falls below expectation after the artificial introduction of an event (at $t = 3 \times 10^4$), and lasts approximately $4 \times 10^4 < t < 6 \times 10^4$. In the case of \mathcal{N} , clearly it remains below the reasonable expectation most of the time, except during the debate episode and during election day, although in the latest the increase is moderate (the same in the range $4 \times 10^4 < t < 6 \times 10^4$ in the model). Similar results are obtained when measuring the average and standard deviation instead of median and IQR.

Figure R2: Evolution of Q and \mathcal{N} in the Spanish dataset (left) and the model's outcome (right). The median of the values is shown as a solid horizontal line, and the shade around it represents the interquartile range. Clearly, modularity (top row) mostly remains in the higher range (and at times well above it), but falls below expectation during the exogenous events (debate and election day for the dataset; $t > 3 \times 10^4$ for the model). On the opposite, nestedness (bottom row) remains low, except for those times in which exceptional events occur, in which the median + IQR is largely surpassed. The model displays this exact behaviour perfectly.

All in all, these additional insights prove that the changes in Q and \mathcal{N} are beyond reasonable expectation, and are thus statistically robust.

These figures and adapted text have been added to the supplementary materials, and we briefly mention the statistical robustness of the descriptors in the main text.

2.2. The first and most striking observation, the increased activity needs to be discussed more. How does this affect the bipartite networks, i.e., how is the increased activity distributed among users and hashtags? (histograms, network plots?) While the two measures are interesting, the networks and their change can be characterized more detailed. Also, how does, for example, the composition of hashtags change during the events?

The referee makes a very timely observation here, from the intuition that the increase in activity may be the driving mechanism of the reported observations (and, specifically, responsible for the increase in nestedness/decrease in modularity, which we also address after comment (3)

below). Therefore, it is important that we disentangle which effects might be explained solely by the increase of activity, and which not.

- Increased activity and structural shifts

We start our analysis with a general observation: an increase in activity may lead to higher network density, but it does not imply an increase in nestedness. To prove this statement we provide a simple example from the Spanish dataset: we take a snapshot of the system right before a large event occurs (*debate*), in which the system is clearly organized as a modular network and connectance (density) is $C = 0.003$. Now, we start simulating an increase in activity (which amounts to an increase in density), taking the system from the initial connectance to a final one of 0.005. We choose this network density, as this is precisely the connectance in the empirical data by the time we observe a maximum in nestedness (during the debate).

Figure R3 summarises this experiment. In the left panel, we see the network at its “starting point”: Q takes a high value (0.677) while nestedness is negligible (0.058). Adding activity at random, up to $C = 0.05$, does not lead to a nested architecture. On the contrary, the process of adding links to the network has deteriorated both Q and \mathcal{N} . The second panel of Fig. R3 shows the evolution of both quantities as links are added, and the third panel shows the resulting network. For the sake of comparison, the panel to the right shows the actual (empirical) network at the end of the process, in which nestedness peaks at 0.58 --an order of magnitude above what one would observe if activity is increased at random. It is clear that increased activity does not render, *per se*, any gains in \mathcal{N} --rather, nestedness stays negligible to a value of in the order of 10^{-2} .

Figure R3: Expected effects of random increased activity. With real data, the system, organized in a modular pattern initially (first panel) transitions to a highly nested one (last panel to the right); however, increasing connectance (activity) in a random manner does not imply *per se* increases in the values of nestedness (middle panels).

We are aware that this result clashes with the idea that connectance underlies the emergence of nestedness (which, in turn, relegates nestedness to a structural byproduct of network density), e.g.:

A. James, J.W. Pitchford and M.J. Plank. Disentangling nestedness from models of ecological complexity. *Nature* 487, pages 227–230 (2012)

Although our aim is not to confirm or debunk these ideas (now cited in the new version, ref. [42]), we remark that the indeed an increase in connectance implies an increase in nestedness *when measured with descriptors that lack a null model*, e.g. Almeida-Neto's NODF (ref. [50] in the original manuscript, ref. [64] in the new one). Instead, in this work we consistently use the measure of nestedness as defined in ref. [26] (ref. [34] in the new one), which already discounts the amount of overlap that two species may have by chance.

This result (included now in the Supplementary Materials in a dedicated section), which is illustrated here for a specific moment in a single dataset, is further generalised, more systematically, in the response to comment (3).

Before moving on to other aspects, there is still an observation that deserves being highlighted. Both datasets in the main text (Fig. 1) present cases in which the transition to a nested architecture happens together with a rise in activity (although overall connectance is not changed by much, see Figure R3 above). This, indeed, may mislead the reader to believe that there is a causal link between these two aspects (activity driving nestedness). However, it is remarkable that nestedness may emerge *without* an increase in activity: see for example the plots in the Supplementary Material (Fig. S4) for the Catalonia, Honk Kong and Charlie Hebdo datasets: shifts to nestedness in these cases are not directly related to increase in activity. The only explanation left --which is the one that the model captures well-- is then a drastic rewiring event, by which user-meme interactions shift preferentially (and temporarily) towards the memes relevant to the rising topic.

We have considered switching figures, so that these possibilities are reported in the main text. However, we feel that the Spanish elections dataset and the Nepal earthquake are good representatives of two kinds of events (expected and unexpected), and thus have left the text as is. Nonetheless, we have introduced clarifications regarding the non-causal relationship between nestedness and activity.

- Increased activity: insights from the users side

Already in the top panels of Fig. 1 of the main text it is clear that some exogenous events cause an attention shift towards a specific topic, and such increased attention can be sensed with large changes in tweet volume. It is unclear, however, whether this change is caused by the same users increasing their activity, or by the arrival of new users who produce a relatively constant average number of tweets/hashtags. As always dealing with real data, the answer is not clearcut, but Figure R4 (left panel: first and second rows) shows that most of the increased activity during extraordinary events is due to new users entering the topic (top-left panel), and that these produce a very large amount of hashtags as well (second row-left panel).

Figure R4. Top-left: evolution of the number of users during the Spanish election cycle. Bottom-left: average number of hashtags per user in the same episode; the shaded area around the average represents the standard deviation above/below the mean. Top-right: Evolution of the number of hashtags in the Spanish dataset, which clearly increases during the identified exceptional events (debate, election day). Middle-right: hashtag coverset at the 99% threshold for the Spanish dataset. Bottom-right: hashtag coverset at the 99% threshold for the model's numerical simulation.

These panels might be interpreted as a proof that the model cannot mimic the observed behaviour in the data, e.g. user's degree (the number of hashtags they use) is highly fluctuating in the data, while it remains constant in the model. However, users' average activity (hashtags per user, $\langle h \rangle$, bottom-left panel) does not change drastically throughout the episode. This finding is very interesting, because it provides a stronger link between actual data and the model (for which $\langle h \rangle$ is constant by design, see panel on the right): when the users' attention profiles are shifted, these do not significantly increase their activity (in terms of hashtag usage) on average, but rather start switching towards the topic on which that same activity is devoted.

- Increased activity: insights from the hashtags side

Here we discuss whether the observation of an increase in the total number of tweets during an exceptional event (Fig. 1 of the main text) is enough to predict a growth in the total number of hashtags in the system: although improbable, it is possible that all new activity could be devoted to a single hashtag. Fig. R4 (right panel) renders a very interesting result with regard to this question.

If we count the number of unique hashtags in each time window of the Spanish dataset (second row-left panel of Fig. R4), it is very clear that the absolute number of hashtags dramatically increases during highlighted events (debate, election day). This may give the impression that the increase in users (top-left panel) brings a noisier environment, with lots of topics being discussed at the same time.

In the third row-left panel of Fig. R4, however, we try to account for the diversity of hashtags in terms of a coverset. In particular, we count the minimum amount of hashtags needed to account for a large fraction of the users in each time window. This implies an iterative scheme: we count

(and remove) all users who tweeted the most frequent hashtag; then we count (and remove) all users who used the second most frequent hashtag; and so on, until we reach a desired threshold (e.g. 99% of the users). Note that, counted in this way, this “hashtag coverset” represents to what extent users are focused on only a few items, despite the presence of many more memes in the information ecosystem that may (or probably may not) get anyone’s attention. It is very interesting to see that for the whole election period (one month of data, approximately), 99% of the users can be accounted for with no more than 400 hashtags, but during intense attention episodes, 100 or less. The behaviour is qualitatively mimicked as well by the model, see the corresponding panel to the right.

With the previous analysis we can conclude that the emergence of nestedness is not bound to an increase of activity neither from the user side nor from the hashtag side, but that for this increment we require a macroscopic reorganization of the interactions within the system. Indeed, we can find (Fig. R3) similarly dense networks with very different values of nestedness and modularity.

Additionally, and more importantly, we observe that the activity per user $\langle h \rangle$ is quite constant across the dataset, and also that during peak events, although the absolute number of used hashtags increases, the number of *crucial* hashtags (coverset) reduces. This means that users tend to focus on a smaller amount of hashtags (generalists memes), the ones which they think serves them best to locate themselves in the ongoing discussion.

Finally, and specific to the question answered here, we have also compared the empirical results with the outcome of the proposed model (right column of Figure R4). These comparative results show that the overall behaviour observed in our real data is maintained on the model. The specific details of this comparative analysis will be commented in the answers below.

(3) Is it true that nestedness and modularity are exclusive in all regimes, how do they relate to network density? Especially, because the Null model here could be that the activity (however it is distributed, e.g., just randomly) is just quantitatively increasing, which could be driving the observations e.g., of reduced modularity. This needs to be tested and the measures used need to be compared to other observables to allow arguing for the modelling approach.

After this comment, and re-visiting closely the text where this point is explained, we realise that the previous version of the manuscript was not clear. We apologise for that.

Nestedness and modularity are not necessarily exclusive, neither in the particular case of online communication networks, nor in general. In a previous paper (ref. [30] in the original manuscript, ref. [41] in the new one) by some of the co-authors of the current work it is analytically proved that there exists an upper bound for the co-existence of nested and modular structures. This bound is $Q \leq 1 - \mathcal{N}$. Therefore, it is true that a highly modular structure can only “afford” a non-nested structure, and the other way around. Note however that this does not impede many other regimes: it is perfectly possible (and actually frequent) that both Q and \mathcal{N} are extremely

low (e.g. Erdős-Rényi networks); or that both have intermediate values (which typically signals the presence of in-block nestedness). This upper bound, and the mentioned mixed scenarios, hold regardless of the size or the density of the network at stake.

With that in mind, it is clear that a reduction in Q (due, for example, to larger density in the network) does not guarantee at all an automatic increase of nestedness. Conversely, a reduction in \mathcal{N} will not imply, necessarily, a larger Q . We have clarified this point in the main text, as it is most important to capture that increases in Q or \mathcal{N} are not expected nor necessary at all. To better grasp this relevant point, the Figure R5 below shows a synthetic example to prove this point. In it, we randomly increase the density of an initially modular network of size $N_{col} = N_{row} = 150$ (first row in the Figure). Each point of the plot corresponds to an increase of 5% in the amount of links. Such random link addition stands, as the referee suggests, as a null model of activity growth: users linking to more and more hashtags at random. Indeed, such random activity decreases the modularity as we move along the x-axis (added links), and yet nestedness remains in the extremely low values that it showed initially. For the sake of completeness, the same process has been applied to initially nested (second row in the Figure), and initially in-block nested (third row in the Figure), structures. The results are very similar: no growth of the complementary pattern is observed at all.

Figure R5: Effects of randomly increasing density on synthetic purely modular (top), nested (middle) and in-block nested (bottom) networks. Clearly, random link addition harms the idealised initial structures, while it has no positive effects on the other descriptors.

As a final note, and completing the response to 2.2, it is clear that a sole increase in activity (in any situation) is not necessarily related to an implicit increment of any of the measures used in the paper. For all three measures, it is required that some explicit driver on the network constituents and their interactions guides the changes at the macroscopic or mesoscopic (in case of modularity and in-block nestedness) levels.

This discussion and accompanying figures have been included as well in the supplementary materials, in a dedicated section.

(4) Also, further observables would allow more quantitative comparisons with the model (e.g., degree distributions etc.), which stay currently qualitative.

Many of the concerns raised by the referee in previous and subsequent comments are related to a deeper, more explicitly quantitative connection between the model and the data. We thank the referee for this, as we believe the outcome in our replies substantially strengthen the value and range of our work.

For example, Figures R1-R4 already provide quantitative ties between the data and the model from different perspectives. Furthermore, in our reply to comment 7, we deal specifically with abundances (both in the data and the model), again by suggestion of the referee. Additionally to these new experiments the previous version of the main text already contains some quantitative connection between the model and the data. Specifically, in Fig. 1 and 3, we monitor the evolution of the relative size of the largest (N_{B1}/N) and second-largest (N_{B2}/N) nested blocks with excellent agreement between them.

Nonetheless, we agree with the referee that a comparison considering basic network statistics may complement in a good manner the existing experiments in the paper's original and current versions. Thus, we added to the supplementary material a comparison between the observed (Spanish dataset) and the model's degree distributions, both at their modular (pre-event) and nested (event) stages, see Figure R6. Notably, the degree distribution in different states of the system shows a good agreement.

Figure R6: Comparative analysis of the hashtag degree distribution. First row is obtained during the pre-event stage (left column) which presents a modular organization both in the data (first row) and the model (second row); and during the event stage (right column), which presents a nested organization.

Finally, we want to remark that further quantitative comparisons are difficult to perform, since our synthetic approach is very simple in purpose: not only our toy model has an arbitrarily small size (although it preserves the scaling of connectance, see Sec. 3.2 and Fig S5 of the Supp Information, original submission), but also we impose a small amount of “topics” (4), which, on top of that, are equally sized (exactly a fourth of the synthetic users and memes are centered around each topic). Clearly all of this represents an idealisation of actual systems, and therefore some specific empirical particularities cannot be matched. Our modelling framework is a simple (but not too simple¹) way to understand the fundamental ingredients driving the observed emergent patterns in online communications systems.

(5) Modelling framework: Here, the connections to the observations need to be made much more explicit. Which quantities are conserved in the model, which in the data (e.g., are the number of connections preserved in the model? Are they in the data?)? Referring to the remark above, as the change in the activity seems to be the dominant observation, it needs to be explicitly addressed in the model.

We thank the reviewer for pointing this out. The evidence to answer these questions is scattered in different places in this letter, which we try to summarize in the following lines.

In the first place, we have seen that an overall increase of activity (more users tweeting on one topic) does not imply that users are producing, individually, more hashtags (see bottom-left panel in Figure R4). The model captures this feature well by design, through the rewiring scheme: although some variability is permitted, users (but not hashtags) mostly keep their degree $\langle h \rangle$ constant.

Regarding the number of links in the network, we note that the empirical ones have varying sizes as the system evolves, as opposed to the synthetic ones that are used in the model. And yet, the latter preserve the scaling of connectance, see Section 3.2 and Figure S5 of the Supp Information, original submission).

On the other hand, we have already discussed the role of changes in activity, which cannot explain at all the structural transitions and elasticity observed (Figures R3 and R5). We also point to our reply to comment (8) below, from a more qualitative point of view.

Finally, there is a last aspect that has not been treated in previous responses, related to the temporal continuity of the topics in the empirical dataset. To this end, we use the most relevant hashtags in the dataset (using the hashtag coverset introduced above), and monitor to what extent each consecutive snapshot contains the same amount of hashtags, i.e. whether these snapshots preserve some sort of semantic coherence. Figure R7 represents the overlap between consecutive hashtag coversets ($|H_t \cap H_{t-1}| / \min(|H_t|, |H_{t-1}|)$), showing a strong

¹ “Make everything as simple as possible, but not simpler”, in Einstein’s words.

persistence of the main hashtags in general, and above 70% during the highlighted events (dips in the overlap correspond to night periods, in which activity remains very low).

Figure R7: Overlap across time between the most relevant hashtags in consecutive snapshots of the Spanish dataset.

(6) Similarly, the number of hashtags is constant in the model, while it changes in the data. How does it change in the data? Can this be incorporated in the model? Again, quantitative comparisons would help to show the connection of the model with the data, which has a sufficient complexity to be more explicit here.

The answer to this comment, like many in this letter, demands the consideration of subtle nuances. On one hand, the direct answer to the question (*Can this be incorporated in the model?*) is “no”: as stated explicitly in the paper, the model does not, at this stage of development, consider birth/invasion processes. This implies that, in synthetic simulations, the number of memes (and users) can only decrease (if its abundance falls below a given threshold). In any case, the incorporation of the mentioned processes is in our short-term agenda.

However, the model does reproduce an interesting fact, which has to do with the attention that certain hashtags receive during extraordinary events. As seen before (third row of Figure R4 and accompanying text), the number of hashtags needed to “cover” 99% of the users decreases when exceptional events arise. This may be interpreted as users who choose to stop producing “less useful” hashtags, and focusing on those that are central to the event. Such behaviour is mimicked as well in the model (see bottom-right panel of Figure R4). This coincidence points to the fact that the model reproduces well the evolution of *effective* hashtags in the system, which indeed does remain constant throughout the evolution of the system.

Part of this discussion has been included in the Supplementary Materials, in the section dedicated to the implications of activity changes on the system.

(7) While the “abundance” stays a theoretical concept without counterpart in the data (likes would be a measure?), other parts of the model can be explicitly compared (e.g., distributions of links, activity distribution, which is currently uniform in the model?).

We thank the referee for the insistence on this point. We also note that this reply is related as well to comments 2 and 3 by referee #2, where the abundance of hashtags is mentioned (comm. 2) along with a reference to the “traces” that might be observed after a shock (comm. 3).

In our original submission, the focus on the macro- and meso-scales had prevented us from even attempting any insights at the microscale. Here we attempt to perform a comparison between the model and the data by exploiting the concept of abundance.

The translation of the concept of abundance to the online communication environment can be thought of as the number of times an item is present on screens. With a language abuse, this is tantamount to the number of individuals (e.g. hashtag *instances*) that build up the species (e.g. *the* hashtag). Following this line of reasoning, for the empirical data on the hashtags' side, we track the hashtag usage frequency over time, as a proxy for hashtag abundance from the model.

We compare the evolution of such abundance in the model and the data (Spanish and Nepal datasets). Top-left plot of Figure R8 shows, from our numerical simulations, the changes in abundances of the hashtags over time, identifying with a colour the topic they are ascribed to. We observe that prior to the event, the abundances of the hashtags are distributed rather uniformly within a narrow range. After the onset of the expected event, however, the abundance of the hashtags in topic 3 (the one to which users' attention is shifted to) begins to increase. In the time range $3 \times 10^4 < t < 5 \times 10^4$ we observe a clear separation between the hashtags from T3 with respect to the ones from the other topics. In our simulations mimicking expected events, the artificial shock peaks at $t = 5 \times 10^4$. Slightly *before* that time, hashtags in topic 3 witness an even stronger increase up to $t = 6 \times 10^4$ (that is, beyond the peak time). After that, the system stabilises and appears to be unable to bounce back to the original, quite uniform abundances.

The top-right plot of Figure R8 shows the usage frequency of actual hashtags over time in the Spanish dataset, where events are known in advance (in this case, election day). Adapting the logic of the model to empirical data, we show the trajectories of a group of hashtags which belong to 4 different communities, the largest ones shortly before (light blue, orange, fucsia) and at the time the ballots were closed (violet). As in the left panel, the vertical lines show the buildup of conversations ahead of the results (around 28 April, 4pm, “event onset” tag), and the electoral schools closing time (28 April, 8pm, “event peak” tag). Overall, we observe a striking qualitative agreement between the simulated hashtags abundance (model) and the hashtag frequencies (data). Until 4pm, all 4 communities present a rather flat and uniform activity (note the logarithmic scale: apparently large fluctuations, e.g. between 12pm and 2pm, imply frequency changes below 10). In the period 4pm-6pm, the behaviour of the violet subset of

hashtags resembles that of the hashtags of T3 when the event occurs (slow but steady separation from the other hashtags, with a frequency increase between 10^1 and 10^2); and also a more pronounced boost in the period 6pm-10pm (i.e. shortly *before* and *after* the event peak). The violet subset of hashtags clearly dominates the scenario even at midnight, and starts an expected decline as conversations mostly halt during the night period. On the other hand, the subset of hashtags from the pre-debate stage (following T1, T2, T4 in the model) present moderate decreases before 4pm, and losses are stronger after that time (especially light blue and orange topics).

Figure R8: Synthetic (left) and empirical (right) hashtag abundance evolution. In both panels, dotted lines represent individual trajectories, and the coloured solid lines represent the average abundances of each topic/community.

For a complete picture we study as well an unexpected event. The bottom panels of Figure R8 represent the evolution of abundances in an artificial setting with an unexpected event happening at $t = 3 \times 10^4$ (left); and the evolution of hashtag frequencies around the time of Nepal's earthquake main aftershock (May 12, around 5pm). Similar to its "expected" counterpart, our numerical experiments on the left show a separation of the violet hashtags in T3, with slight decreases of the other topics T1, T2 and T4. The system also appears to be unable to return to the pre-event stage, and so the only obvious difference is that the separation occurs in an abrupt way. On the right, we see the evolution of the frequencies of hashtags that belong to four of the largest communities detected in the data, slightly before (light blue, orange,

fucsia) and right after the aftershock (violet). Clearly, hashtags in the violet community present a sudden increase, followed by a very slow decrease resembling the one observed for $t > 6 \times 10^4$ in the left panel. Given the international impact of the earthquake in Nepal, there is not a decay during the night period.

These two examples uncover new facets of our model that had remained unexplored in the original manuscript. On one hand, Figure R8 extends the meso- and macroscale connections between data and model to the microscale. On the other, it provides a different perspective of our approach with regard to the memory of the system, and the trace that exceptional events leave behind. From the meso- and macroscale, it is still valid to say that the system is trapped in a narrow set of structural configurations (namely, nested arrangements with only one or several blocks): this explains our use of the term “elastic”. And yet, structural elasticity does not imply that the dynamical states of the system remain the same. Strong enough perturbations push the system away from its present stable state towards a new one. We intend, in the short-term future, to further study further how strong the perturbation should be to jump to a different stable state.

From the users perspective the issue becomes less obvious. Applied to users, the *rationale* above (“the number of times an item is present on screens”) implies that visibility should be quantified as how often a user’s profile appears on others timelines, but that is not something that can be collected from online services’ APIs. At the system level and long time scales, visibility might be correlated with the number of followers, but this quantity can be regarded as constant at the short time scales that we consider in this work, and that the model aims to grasp. Therefore, quantifying visibility at the microscale is elusive, although some nuanced proxies may be possible --we have recently released a pre-print along similar lines:

CA Plata, E Pigani, S Azaele, V Callejas, MJ Palazzi, A Solé-Ribalta, S Meloni, J Borge-Holthoefer and S Suweis. Neutral Theory for competing attention in social networks. arXiv:2006.07586 (2020)

We judge these new micro-level insights very valuable. Thus, we have included them in the main text. Figure R8 appears now as Figure 4.

(8) Relatedly, when the external shocks are incorporated the distinction between expected and sudden events is made, but are they distinguished in the empirical data. In Fig. 1 the two types are shown but the reaction of the system does seem to be most evident in the activity, rather than the other observables.

(comment (4) by referee #3 points at a similar question; we provide here a longer answer).

The referee is right on this observation, which came to a surprise for us as well. At the research planning stage, we wanted to make sure that the collection of empirical datasets was not restricted to a single topic (politics, sports, etc.), and also that the analysed episodes would not have the same temporal profiles (thus our resorting on ref. [39] in the original version, ref. [36] in the new one). Also on the model's side, we ensured that attentional shift functions could be designed with *ad hoc* temporal profiles and plugged into the evolution of the system at runtime (e.g. Figure S3 in the original Supplementary Materials).

In considering these distinctions, we intended to analyze whether the system's structural response to environmental shocks (events) was different. However, we observe that transitions to and from nested patterns (or, to be precise, to and from one-block, in-block nested patterns) are quite stable, regardless of the type of event. If any, the observed differences are in the abruptness (speed) at which the structural transitions take place, which can be appreciated in the lower panels (increased temporal resolution) of Figure 1 of the main text. As in other facets, also here the empirical findings and the model's results are in good agreement, i.e. no significant differences, except for the abruptness of modularity's breakdown, can be observed). And yet, we have not worked in the precise adjustment of the parameters that rule external events in the model.

This is used to model the attentional shift function in the supplementary material (the most explicit relation to the data), but compares different observables, right? But an increase in inactivity does not necessarily reflect a shift of attention.

The referee rightly points at a subtle issue here, that demands clarification. Indeed, the confidence of an activity increase as a (naive) predictor of an attentional shift should be limited. To be clear: in general, we cannot establish a direct relationship between changes in activity, and changes in attention. However, we use the increase in activity as an indicator of the duration/response to the external event (following ref. [39], ref. [36] in the new version). As we have noted in previous responses (also in some of the datasets included in the paper, e.g. Hong Kong protests), it may well be that a shift in attention is not accompanied by significant changes in tweets volume; and vice versa, increases in activity are not necessarily caused by attentional shifts (Figures R3 and R5).

This general rule has a logical exception, in the case of unexpected events (e.g. an earthquake, or the demise of a celebrity). Because of their very nature, the activity for those events is null

before they occur, so any attention shifted towards such events is necessarily linked to an increased activity.

Referring to an earlier point, to test the null model of randomly increasing activity, this can be checked in the data to back up this assumption.

Yes, thanks. Figures R3 and R5 above, and the accompanying explanations, already tackle this comment.

(9) The in-block nestedness is an interesting measure, but seems to be quite sensitive to the community detection method. That should be addressed.

The concept of in-block nestedness has been around for over a decade (starting with ref. [25] in the original manuscript, ref. [33] in the new one) and under different names, but a dedicated method to quantify and detect it was not developed until 2018 (ref. [26] in the original manuscript, ref. [34] in the new one). This implies that not many optimisation methods are available so far.

From a computational point of view, searching for a (sub)optimal in-block nested partition is very similar to searching for a modular one: the choice of an maximisation heuristic tries to balance computational speed with accuracy. We are fairly sure that our choice (Extremal Optimisation) makes a good job at finding good in-block nested partitions, for several reasons:

- (1) The algorithm performs excellently when confronted with a synthetic toy model, for which the ground-truth partition is known in advance (for several regimes of network size, intra- and inter-modular noise, density, etc.). This is tested in depth in ref. [30] of the original manuscript, ref. [41] in the new one.
- (2) The Extremal Optimization is known to provide a very convenient balance between accuracy and costs, and this especially true for small to middle-sized networks (from hundreds to few thousand nodes). Ref. [22] in the original manuscript (ref. [30] in the new one) reviews this point, and also the reference

Danon, Leon, et al. "Comparing community structure identification." *Journal of Statistical Mechanics: Theory and Experiment* 2005.09 (2005): P09008.

which we have now introduced in the Supplementary Materials.

- (3) Finally, we compare our optimisation results to an alternative heuristics (bee swarm optimisation algorithm, used in ref. [26] in the original manuscript, ref. [34] in the new one), which is known to deliver high-accuracy results at the cost of a large computational effort. Precisely because of this high cost, we cannot present a comparison based on the Twitter networks of the manuscript, but rather on a collection of smaller networks. Particularly, we compare the two algorithms for a collection of 140 ecological networks. The following plot shows a scatter of the obtained values of \mathcal{I} after each optimization

process. The solid blue line corresponds to the linear fit, and the solid orange line corresponds to $y=x$. We computed the Pearson correlation coefficient (Pearson=0.806) and highlighted in red the networks for which the difference between the two values is less than 0.05. In general, we observed a good agreement, and a high correlation between the \mathcal{I} values obtained from each strategy. As expected, since the bee swarm algorithm is more exhaustive, it usually delivers higher \mathcal{I} values. Nonetheless, we can observe that the bee swarm heuristic performs better than EO in most small networks, but worse for the two largest networks compared (in the order of hundreds of nodes).

Figure R9: Scatter plot comparing the \mathcal{I} values for two optimization strategies.

(10) *Minor remarks:*

- *The in-block nestedness needs to be introduced earlier*

We agree with the referee that in-block nestedness is a central piece of the paper and deserves being introduced earlier. However, we believe that there is value in explaining that we only considered in-block nestedness *after* we realised that it emerged from the model. We mean to follow the traditional logic of scientific discovery, i.e. observation, proposal of a theoretical model that explains the data, prediction of new phenomena as implied by theory, and back to observation. Still, the concept is initially mentioned in the Introduction, and briefly at the end of the empirical analysis.

- *A table or list, introducing/listing the variables/ingredients of the model and their counterparts in the data would help a lot*

We thank the referee for suggesting this. We have made large efforts to clarify the core concepts of the paper, also after considering similar comments from other referees. The main consequence of this is a significant enlargement of the Methods section, but also minor clarifications throughout the text (and of course a major revision of the Supplementary Materials). We hope that all this is helpful to see how data and model connect.

- *The caption of Figure 2 is not self-explanatory*

We have improved the caption towards a more explicit one.

- *In the main manuscript (without the supplementary material) it is difficult to follow the structure of the model, better to pull some of it in the main text or in the methods section.*

We have introduced in the main text some of the extended materials that were relegated to the supplementary information, regarding the model and the structural measures used in the work.

- *How long were the snapshots to create the empirical networks?*

This is explicitly described in the Supplementary Material, but we failed to include the information in the main text, we apologise for that. For the datasets spanning days/weeks, the snapshots are 3 hours of duration. When the work reports on focused events, we have used fifteen minutes snapshots. This information is now mentioned in the main text as well.

- *Typo in page 7 "...adjacency matrix. , and h is the..."*

Thank you. We have corrected the typo, and revised the whole text for other possible ones.

Reviewer #2:

The authors study the structure and dynamics of bipartite network of hashtags and twitter users. I found the approach fascinating and apologize for taking a bit long to produce my review. It simply took me quite some time to get a feel for what is really the result, and how it is obtained. My opinion should not be weighted too heavily as I am not a network scientist. So, please take my views as observations that may hopefully be of some guidance for revising this for the current or another journal.

We thank these positive comments on our work.

(we have numbered the comments for easier traceability)

(1) The title promises an insight in resilience and elasticity, and later the authors mention 'remarkable structural elasticity'. However, it does not seem surprising to me that people start talking immediately about a big event when it happened (elasticity), and then return to their normal interests (resilience).

We thank the referee for this comment which may help substantially to the improvement of the discourse throughout the paper. Following other remarks by the rest of the referees, we have changed the title of the work.

Regarding the “surprise” effect, indeed one expects that people will switch their attention (and orientate their activity accordingly) when special events occur. However, it is not straightforward that the switch in activity will happen in a precise and predictable way. Figures R3 and R5 above show that the transition from a modular to a nested architecture, or from nested and back to modular, is anything but casual. Thus our objectives are rather:

- 1) To point out the existence of ubiquitous/universal emergent patterns in information ecosystems, i.e. the modular structure in user-meme interactions in the absence of big external events, and the nested structures after the occurrence of such events.
- 2) To isolate which is the main driver of information ecosystems, which turns out to be the maximisation of user's and meme visibility.
- 3) To understand how information ecosystems respond to external perturbations, i.e. if and how they are stable and rigid, or flexible systems.

1.1 In nature, bipartite networks of species involve adaptations such as evolutionary tuning between pollinator and specific flower shapes. That takes time, and is quite an investment. Using a hashtag and then dropping it is obviously swifter. In short, I do not see an eye-opener when it comes to the terms in the title.

We agree that the title did not reflect the most important results of our work, and have changed it accordingly. Thanks to the referee's suggestions, we have now tried to improve both the title and the abstract to make the main contributions of the work more clear.

With respect to the reviewer's comment, let us also note that bipartite networks are also used in ecology to study ecosystems at shorter time scales (stability, sustained biodiversity, colonization and invasion processes), and without considering any aspect of evolutive processes. We used our bipartite networks in this spirit. Users tend to choose (and adapt to external events) the most appropriate hashtags to maximise the proliferation of their tweets (abundances), like a pollinator may choose among the available (and feasible) flowers considering the nectar volume, color and abundance. In this sense, one can for instance study how the relative species abundance evolves at a yearly scale and without considering any evolutive process.

Still, and also following the concerns raised by referee #4, we have adjusted our use of certain terminology to avoid inaccuracies and over-claims, dropping the term “coevolution” except when explicitly linked to the interplay of structure and dynamics in networks (along with the references, refs. [22-25] in the new version).

(2) Also, I am not surprised by the nestedness vs modularity results. Isn't it a bit obvious that people from different groups and interests start talking about the same big thing when it happened? This would be visible in all kinds of indicators including nestedness, but also other indicators such as abundance of the most mentioned hashtag, Shannon diversity of hashtags, or network connectivity measures? Thus, what is really the surprise here?

This relevant comment is in close connection with the remarks by the first referee, regarding the relationship between the emergence of nestedness and the increase in activity. We thank both referees for pointing this out, as indeed the issue demands clarification. As stated above, a preferential attention to a given topic might imply an increased activity in that topic, but there is no necessary connection between this and the increase of nestedness.

In particular, it is not possible to translate automatically an increase in activity with the emergence of nested patterns. Users could gather around a single topic in a random way (or any other pattern that one could imagine, for that matter). And yet, among all possible configurations (including a random one), the system self-organises to an in-block nested or fully nested hierarchy. Thus, it is necessary to identify which are the local mechanisms that make the bipartite network to organize from a modular to a nested structure and back. In these lines, our work shows that the definition of communication niches and individual maximization of visibility are the key elements to understand how the meso and macro organizations emerge. We refer to the answers and figures that correspond to comments (2) and (3) of referee #1.

Finally, we may agree that there may be other signatures that can indicate some major change in the system. Hashtag coversets, in the third row of Figure R4 (left: empirical; right: model) point in this direction. These coversets, or measures like the Shannon diversity, are some among many ways to quantify the emergence of large heterogeneities among hashtags. However, again, a decrease in the diversity of hashtags does not imply nestedness, i.e. it is a *necessary* but not *sufficient* condition for the presence of a nested architecture.

These ideas are included now in the Supplementary Materials in a new dedicated section 3.3.

(3) This may sound a bit negative, but I do think the information the authors analyze is fascinating. I would just think that other research questions could perhaps be more promising. If big events do get everyone talking about the same thing, does this leave a trace in social cohesion? Are groups becoming connected in a meaningful lasting way? To me that would be a deep impact. So the fact that you do not seem to see that could imply that the nestedness and enhanced connectivity is not really meaningful? So big events do not really connect? Or are some kinds of events more connecting than others? Could you detect that?

We appreciate this comment by the referee. Together with comment 7 by referee #1 above, it has triggered our efforts to differentiate the narrow set of structural arrangements that the system is headed to (elasticity), from the lasting dynamical effects, which seem to be pushing the system from one stable state to the next, provided that the external shock is strong enough. Particularly, Figure R8 and surrounding text above (now included as a section in the main text) provide a link between the theoretical concept of abundance and an actual measure such as hashtag frequency, enabling us to compare the evolution of such abundance in the model and the data. Again, we thank the referee for insisting on this point, as it has pushed us to explore a facet of the work that, we believe, is now stronger. We refer to our response to comment 7 above for a detailed discussion.

From the users' perspective, there are many open (and exciting) questions lying ahead. As a matter of fact, we explicitly mention, in the Discussion section, that this work rules out interesting phenomena like "cultural drift": the fact that users have memory and, as such, their "interest niche" should change in time, i.e. big events should leave a trace in their connectivity. However, tackling this particular problem demands a much longer time scale to observe how the drift towards new interests consolidates, and accordingly a much broader data accessibility (note that, for example, our datasets do not track following/follower relationships, which could be a hint to a user's changing interests), which now lie beyond our current capabilities.

Reviewer #3:

(we have numbered the comments for easier traceability)

(1) I am surprised to see several new concepts frequently used in this paper including its title, but the authors often choose these terms without specific and valid explanations. While the term "resilience" has often been used across scientific fields, the term "elasticity" is not the concept with which (social) scientists are familiar.

We are sorry to see that some of the key concepts in the work are confusing.

Admittedly, resilience is a debated concept. Its use and abuse in many fields (psychology, ecology, engineering, etc.) has led to a certain vagueness of proposed definitions, which in turn can lead to misunderstandings and impede its application to systems modelling. In our case, we have tried to be specific about what we understand by system's resilience, in the paragraph (page 3 of the original submission):

“System resilience or stability is defined in different ways in ecology and environmental science, but can generally be thought as the ability of the system to recover the original system's state after a perturbation of the model state variables or parameters. Specifically, in the case of structural elasticity, the system state is not given by the nodes' configuration (e.g. the abundance of each species), but by the overall network architecture (i.e. modular, nested), which is perturbed by the external event.”

The paragraph comes with 6 citations (deleted here for easier reading), 4 of which sustain our use of the term. Accepting this specific sense of resilience, which implies recovery to an original state, then the near-assimilation to elasticity is natural: the system is resilient because it shows elasticity (in an almost mechanical sense).

Having said that, we were unaware (and have recently realised, thanks to the referee's comment) that elasticity has a very different meaning in social science, and particularly in economics. Indeed, if one reads “elasticity” in this work as it is understood in economics, then the term impedes, rather than improves, the understanding of the paper.

To avoid this misunderstanding, we have added a clarification in the introduction, so as to settle our use of ‘elasticity’. The text now reads (in a different colour here and in the manuscript):

“Here, we provide evidence that information ecosystems exhibit a remarkable **structural elasticity** to environmental changes, **recovering its original architecture in the aftermath of an external shock.**”

(2) As a matter of fact, the first paragraph starts with several undefined terms and/or phrases including bio-cognitive limits, quotidian communication processes, oligopolistic media environment, attentional resources of the audience, etc. I am not criticizing the authors' language skills in English, but pointing out what core concepts are and what the surrounding terms are. Relations among conceptual words should have been revealed in a clearer way.

We hope that the aforementioned clarifications help to frame some of the key concepts.

Regarding the specific expressions mentioned here, we have avoided explicit definitions because some of them are widely known and have been thoroughly discussed in the literature in the last years. A dedicated definition would block the flow of how the manuscript is laid out. This applies, for example, to the bio-cognitive limits of individuals (and the related “attentional resources of the audience” in information-rich environments), for which we cite classical works in economy (Simon, Kahneman) and cognitive science (Posner).

We have changed the expression “quotidian communication processes” to “everyday communication processes” --but no citation or definition is given, as it refers literally to our daily experience of communication (which, in the pre-digital age, did not imply normally any pressure to our attentional capacity).

For the case of “oligopolistic media environment”, again this is just a descriptive expression. The term “oligopolistic” is commonly used with a negative political charge, but here it is used in its literal (and even etymological) use: “a situation in which only a small number of companies are involved in producing a particular type of goods or in providing a particular type of service” (from the Cambridge University Press dictionary). Thus, we don't think it demands a citation or a justification: it is a fact that, prior to the web 2.0 (or, if one wishes, prior to the email), our exposure to information sources (media) was extremely limited ($S \ll R$, where S is the number of sources and R the number of receivers), as very few actors had broadcasting capabilities.

More generally, we have added the following citations (refs. [5-7] in the new version):

Anderson, S., A. de Palma. 2012. Competition for attention in the information (overload) age. *Rand Journal of Economics* 43(1) 1–25.

Van Zandt, Timothy. 2004. Information overload in a network of targeted communication. *Rand Journal of Economics* 35(3) 542–560.

Iyer, Ganesh, and Zsolt Katona. "Competing for attention in social communication markets." *Management Science* 62.8 (2016): 2304-2320.

Although different in the objectives of the paper, these works introduce (and explain) some ecological terminology used in the context of online communication, and thus helps us to bridge the gap between potentially diverse readership.

(3) Instead of "resilience" used in the title, I suggest that the authors choose "modular" VS "nested" architecture as the core concept of this paper. Thus, introduction should begin with them at the center of the writing.

Following the suggestion of the referee, we have changed the title.

(4) Public attention toward elections increases over time, but when an earthquake occurs, social interest explodes and disappears immediately. The choice of two datasets will significantly affect the findings.

We thank the referee for the comment. The idea that different events have different temporal profiles is also present in the paper, both explicitly and through ref. [39] (in the original submission). Our choice of datasets responds precisely to our intention to show that the observed structural signatures are robust to the topic, and to the (un)expected nature of the event, etc. This is valid for the examples in the main text (Spanish elections as a planned event, against the Nepal earthquake aftermath), but also for the ones in the Supplementary Materials: the FIFA dataset has certain planned "periodicities" (games every few days, over a whole month), but it is not known in advance which teams will play the final stages.

If any, the differences must be sought in the abruptness (speed) at which the structural transitions take place --but this aspect is beyond the scope of the current work.

See also our response to comment (8) of referee #1 above, which is related to the predictable/unpredictable nature of events.

Last but not least, Han Woo Park's team has conducted numerous social media research in election and/or environmental disaster contexts including a recent Covid-19. You might like to take a look at and refer to their research.

We thank the referee for pointing out this work, which we were not aware of. Given the proximity to the topics in our work, we have added the following reference to our manuscript (refs. [38] in the new version):

Park HW, Park S, Chong M. Conversations and Medical News Frames on Twitter: Infodemiological Study on COVID-19 in South Korea. *J Med Internet Res* 2020;22(5):e18897. DOI: 10.2196/18897

Reviewer #4:

First of all I would like to apologize for my long delay in sending my review. This manuscript fits the long-lasting quest searching for general patterns in complex systems. Thus, the questions addressed are timing and the patterns reported and the approach used very interesting.

We thank these positive comments on our work.

Having said that, as an ecologist and evolutionary biologist, my main concern is on the comparisons with ecosystems the authors used. In short, I think the ecological analogies used at the best inaccurately described and maybe flawed, imperiling the implications of the study to ecology and to the synthesis between theoretical ecology and applications in other scientific fields.

Main comments

(1) The underlying reasoning of the manuscript is based on the idea of actor-meme coevolution. However, as far as I understood, memes are not shifting their patterns of interaction to maximize any quantity. They are passive resources that users interact with. So, coevolution - in a biological sense, see my comment #3 - does not seem to me a valid benchmark. At the best, it is analogous to evolutionary resources against fixed resources and how it is modulated by other environmental factors.

We thank the referee for this insightful comment (also, the reference to the Gaia hypothesis later on), which we judge as extremely important: we do not aim to merely present an appealing analogy or an easy metaphor. We explicitly intend to stay away from a frivolous misuse of the ecological framework, and we apologise if the manuscript contains conceptual and/or terminological inaccuracies (“coevolution”, “mutualistic ecosystems”, etc.). We have solved them, as stated in the response to the corresponding comments 2.2 and 3 below.

On the other hand, the reviewer points at the consumer-resource paradigm as a more suitable frame for our manuscript. However, we respectfully disagree. In fact, the only condition that defines mutualism is the exchange of goods or services between two species, i.e. the fact that each species involved in mutualism must receive a benefit from the interaction. The fact that users actively select the memes to obtain visibility, and memes passively gain outreach from the users they interact with, we believe, suffices to describe the system as mutualistic.

Nevertheless, we agree with the reviewer that memes do not optimize any quantity, and this was probably not clear enough in the previous version of our manuscript. In fact, in our simulations, only the users are the ones that can actively “rewire” their links to new memes. The rearrangement of the memes network structure is simply a byproduct of the users actions. We now make this point clear in the revised work.

Finally, we would like to highlight that we do not claim that user-meme dynamics need be identical to the corresponding ecological one. We do claim that information ecosystems and

natural mutualistic assemblages might belong to a broader class of bipartite systems, i.e. those dominated by intra-class competition and inter-class mutualistic interactions (see the end of response to comment 2.2 below).

(2) *Introduction: “Under the light of these four drivers –competition, mutualism, coevolution, environment–, online communication systems become a special case of mutualistic ecosystems”. This is an inaccurate statement.*

2.1. *First, the citation here is misleading since the paper cited does not address online communication systems but the network structure of mutualistic assemblages.*

We agree that the citation here does not help to clarify the statement, so we have removed it.

2.2. *Second, there is no such thing as “mutualistic ecosystems”. Mutualistic interactions are ecological interactions in which individuals of different species benefit from the interaction. Mutualistic interactions may lead to evolutionary or demographic effects at the population level. In some types of interaction, mutualisms may involve individuals from multiple species, forming mutualistic assemblages. Ecosystems, in turn, are systems in which organisms and environment are connected by pathways in which matter and energy flow. I have to say that this lack of accuracy with biological terms is not expected in a study that aims to connect different fields and similar misuse of ecological terms led, in the past, to multiple research dead-ends and simply wrong ideas that create more confusion than light (e.g., Gaia hypothesis).*

We understand the referee’s point: actually, unlike the case of elasticity (comment 1 of referee #3), we are aware that the expression “mutualistic ecosystem” is a language abuse. At the time of writing the manuscript, we preferred “ecosystem” to “community”, precisely because “community” is also ambiguous --it has a very precise meaning in the network literature (in the sense of “module”; “compartment” in Ecology) and is thoroughly used in this work (for obvious reasons). In this dilemma, we opted to keep “mutualistic ecosystem”, following the *rationale* that, typically, when you neglect predation (and thus consider only intra-class competition and inter-class mutualistic interactions) then you might call them mutualistic ecosystems --but that of course is an abstraction. Also, we felt that our choice was not exceedingly abusive, as the expression has made its way already in the scientific literature, e.g.:

Ellie Nagaishi and Kazuhiro Takemoto. Network resilience of mutualistic ecosystems and environmental changes: an empirical study. *R. Soc. Open Sci.* 5180706 (2018)

Flaviano Morone, Gino Del Ferraro & Hernán A. Makse. The *k*-core as a predictor of structural collapse in mutualistic ecosystems. *Nature Physics* 15, pages 95–102 (2019)

The arguments above explain why we had indulged in the expression “mutualistic ecosystems”. However, we understand and appreciate the referee’s objection towards language accuracy

(specially in interdisciplinary research), and thus we now use “mutualistic assemblage” (avoiding the potentially confusing use of “community”), and “mutualistic network” a few times.

Finally, there are multiple differences between online communication systems and mutualistic assemblages related to spatial, temporal, and organizational scales. In the best-case scenario these two systems share similarities that may drive the dynamics of a given state variable but I do not see how online communication systems are a special case of mutualistic assemblages.

This is a very interesting discussion indeed. Spelling out the differences between, e.g., plant-pollinator communities, and online communication systems is unnecessary --those differences are too obvious, as the referee rightfully points out. Then again, it is a fact that we have observed large advances in some scientific fields when researchers borrow concepts and terminology from other disciplines. In the current days, we are bombarded with expressions like “infodemics” or “infoxication”. Similarly, we say that some online contents go “viral”. Underlying these informal uses there is a large literature, very formal indeed, that models information dissemination in the same way as an SIR (susceptible-infected-recovered) or similar epidemiological dynamics. Resorting to such a powerful framework has given notable results, despite the accepted language abuse when we talk about the “contagion” of information.

Of course, both processes (disease and information spreading) are different, regarding their spatial, temporal and organizational scales (not to mention the particularities of infection mechanisms). And yet, epidemiological models (which precede by large online communication) are useful to unveil and understand certain properties of information systems. It is undeniable that, abstracting the details, has upgraded a useful simple analogy to the modelling level.

Having said all that, we agree with the referee that the sentence is not accurate. We should not say

“online communication systems are a special case of mutualistic assemblages.”

but rather

“online communication systems and natural mutualistic assemblages become special cases of a broader class of mutualistic bipartite systems, i.e. those dominated by intra-class competition and inter-class mutually beneficial interactions, although clearly functioning at very different spatial and temporal scales”.

Also, we have clarified in which sense, epistemologically, we are exploiting the similarities between natural mutualistic assemblages and “information ecosystems”, along the lines of our argument above (comment 1).

(3) The use of the term coevolution. Coevolution here is used, if I got correctly, in the sense of the feedback between the dynamics of networks (structural change across time) and the

dynamics on networks (flow of information), i.e., adaptive or coevolutionary networks sensu Gross and collaborators. Coevolution, in evolutionary biology, however, is the reciprocal change of traits driven by natural selection in interacting populations of different species. In this context, coevolution can occur – and indeed is usually reported – in pairwise interactions. Again, the lack of accuracy when using the biological terms may imperil the analogies drawn between communication and biological systems. For instance, coevolution – in its biological sense - is just one of the processes that lead to this feedback between the dynamics of networks and the dynamics on networks in ecological assemblages. Non-evolutionary, behavioral changes are well known to lead changes in ecological interactions, as exemplified by the theoretical work by Peter Abrams.

We thank the referee for raising this polemic point. Again, we confront here a problem of specialised terminology. As the referee points out, “coevolution” is frequently used in the complex network literature which refers to “the coevolution of topologies and states”, that is, how the topology/structure of the network affects the states of the dynamics happening on top of it; and the other way around, how the states of the units in the system end up modifying the topology of the network. Some literature with this leaning:

Sheykhalı, S., Fernández-Gracia, J., Traveset, A. *et al.* Robustness to extinction and plasticity derived from mutualistic bipartite ecological networks. *Sci Rep* **10**, 9783 (2020). <https://doi.org/10.1038/s41598-020-66131-5>

Gross, T., D’Lima, C. J. D. & Blasius, B. Epidemic dynamics on an adaptive network. *Physical Review Letters* **96**, 208701 (2006).

Vazquez, F., Eguíluz, V. M. & San Miguel, M. Generic absorbing transition in coevolution dynamics. *Physical Review Letters* **100**, 108702 (2008).

Holme, P. & Newman, M. E. Nonequilibrium phase transition in the coevolution of networks and opinions. *Physical Review E* **74**, 056108 (2006).

This is the concept that we kept in mind when writing the manuscript. Considering the referee’s comment, and to avoid further confusion --the ambition to bring together communication and biological systems is already challenging enough-- we resort on the concept of co-adaptation. We feel that this concept reflects better the mechanisms that we describe, and also the paper becomes more coherent with previous work by some of the co-authors. Indeed, ref. [13] (in the original version; ref. [16] in the new one) *never* mentions co-evolution to define his model --they rather use the expression “adaptive evolutionary framework”. Still, we use the term “co-evolutionary” when explicitly linked to the interplay of structure and dynamics in networks (along with the above references, refs. [22-25] in the new version).

(4) *Results: the notion that nestedness and modularity can coexist in modules with internal nestedness is not new. More than 10 years ago, Thomas Lewinsohn and his coauthors*

proposed these combined network structures. There are multiple empirical and theoretical studies exploring these patterns in ecological systems.

We are surprised by the referee's comment on this aspect, who seems to have overlooked that Lewinsohn's paper is indeed cited (ref. [25] in the original submission, ref. [33] in the new one). We believe that we have fairly acknowledged Lewinsohn's work, as well as others in which the concept of in-block nestedness has appeared (under different names): see refs. [44, 45] by Flores and co-authors (refs. [55,56] in the new one). For the sake of completeness, we have also added the references (refs. [57,58] in the new version):

Beckett, Stephen J., and Hywel TP Williams. "Coevolutionary diversification creates nested-modular structure in phage–bacteria interaction networks." *Interface focus* 3.6 20130033 (2013).

Marco A. R. Mello, et al. Insights into the assembly rules of a continent-wide multilayer network. *Nature Ecology & Evolution* 3, pages 1525–1532 (2019).

Which are, besides refs. [25,44,45], the only work that addresses in-block nestedness explicitly, to the best of our knowledge.

The main difference between Lewinsohn's work and ours is that the former is limited to present this possible arrangement, but never provides a clear way to quantify it. Even the works by Flores *et al.*, Beckett, and Mello et al., which explicitly search for intra-modular nestedness in actual datasets, do not provide a dedicated method to characterise in-block nestedness: they rather exploit existing methods, i.e. optimising modularity first, and study nestedness in the resulting partition (a method that may overlook in-block nested structures).

Needless to say, none of those works attempt to develop a generative model to understand the main drivers of the system dynamics leading to the emergence of such hybrid patterns. In the rewriting of the manuscript we have tried to better convey and clarify the original contributions and novelty of our work.

Minor points

1. In addition to inaccurate use of biological concepts, the authors imperil accuracy and precision at multiple parts of manuscript. A non-exhaustive list of examples include but are not limited to:

a. Abstract: *“apparently chaotic and noisy”*

We have changed “chaotic” for “frenetic”, to avoid a possible confusion with a literal interpretation (chaotic dynamics in Physics).

b. Introduction: *entire paragraphs without any reference supporting statements (see minor comment #3).*

Response in minor comment 3 below.

c. Results: *“Biased as it may be, Twitter is without a doubt a sensitive platform that mirrors, practically without delay, exogenous events occurring in offline environments”.*

We have added two references (refs. [37,38] in the new version) supporting the idea that events happening outside Twitter are reflected very rapidly in it

Borge-Holthoefer, J., et al. The dynamics of information-driven coordination phenomena: A transfer entropy analysis. *Science Advances*, 2(4), p. e1501158 (2016).

Park HW, Park S, Chong M. Conversations and Medical News Frames on Twitter: Infodemiological Study on COVID-19 in South Korea. *J Med Internet Res* 2020;22(5):e18897. DOI: 10.2196/18897

along with ref. [39] (in the original submission). Besides that, we don't see any inaccuracy in the sentence.

“vision of the non-virtual world”

We have changed this expression for a simpler one:

“reflection of the real world”

“highly fluctuating nature of this endless communication stream”.

We have rephrased the sentence to a simpler (less literary) version:

“highly fluctuating nature of this communication stream”.

Besides these particular expressions, we have paid attention throughout the manuscript to avoid ambiguous or inaccurate expressions.

2. *Abstract: “selective pressure, i.e. the chances to persist and reach widely are tightly subject to changes in the communication environment” → although selection may change across environment, selection may act even without environmental change.*

We have changed

“In turn, contents are driven by selective pressure, i.e. the chances to persist and reach widely are tightly subject to changes in the communication environment.”

to

“In turn, contents are driven by the evolving context, i.e. the chances to persist and reach widely are tightly subject to changes in the communication environment.”

3. *Page 1: second and third paragraphs are essential to the paper but no references support the claims made.*

We have added two references in the second paragraph (refs. [20,21] in the new version),

Ma, Z et al. On predicting the popularity of newly emerging hashtags in Twitter. *Journal of the American Society for Information Science and Technology* 64(7) (2013)

Cheng, J., Adamic, L., Dow, P. A., Kleinberg, J. M. & Leskovec, J. Can cascades be predicted? *Proc 23rd international conference on World Wide Web*, 925–936 (2014).

which point at the importance of content to predict virality in a piece of information. The focus on content (among other features) is an indirect proof of the claim in the paragraph, i.e. the benefit of choosing the “right” meme is visibility (and the benefit for the meme is a larger spread).

The third paragraph was unreferenced because it does not contain a claim, but rather an observation: users need to adapt their contents to a changing context; also, the success of certain memes that provided large visibility in the past is no guarantee of the same success in the future. Still, we have added a reference (that was used in other parts of the work: ref. [14] in the original version, ref. [18] in the new one).

Reviewers' Comments:

Reviewer #1:

Remarks to the Author:

In the revised version of the manuscript the authors could clarify basically all my concerns with an impressive amount of extra analysis. Even though most links to the data remain qualitative, all the comparisons are reassuring that the empirical observations can be linked to the proposed mechanisms, which seem to describe an interesting and potentially quite general behaviour during shock events on social media.

In particular I am convinced that merely an increase in activity can result in the observations, as the ensemble simulations in the response have shown.

I am convinced that this work presents a robust interpretation, through an innovative model, for an interesting qualitatively similar response of the social media ecosystem across different topics.

Reviewer #3:

Remarks to the Author:

The authors have addressed my comments in a successful way. I am satisfied with their revision in terms of social sciences. Congratulations!

Reviewer #5:

Remarks to the Author:

I have read the manuscript by Palazzi and coauthors with considerable interest. I think that their analyses are quite novel and potentially very interesting, and on the whole appear to be properly conducted. This notwithstanding, I am in very strong agreement with Reviewers 2 & 4 from the previous round of review that the links to the ecological literature need considerable improvement before I could regard the manuscript as publishable. In many ways, I feel that this is a direct consequence of the manner in which the authors have decided to present the study; that is, if they have found themselves between a rock and a hard place after peer review, the current presentation of the study is as much to blame as any use or abuse of language.

I agree with the authors that exchange between disciplines can lead to very fruitful outcomes and hence cross pollination should not be discouraged (all puns intended). However, it is important to get things right (or as right as possible) so that subsequent researchers have firm ground to stand on, and to avoid re-inventing the wheel. Since these authors emphasise "an ecological approach" in their title, it behoves them to get the language right---not from their own perspective but from the perspective of domain experts. Right now, I do not feel their manuscript accomplishes this feat, and the apparent resistance to change (adaptation?) that comes across in their response to reviewers does them few favours. Assuming that everyone can get on the same page in terms of the underlying language (see below), I also have some specific suggestions that might at least "soften" the analogy to ecology to the point that it becomes acceptable (or at least tolerable).

1) Resilience, elasticity, structural elasticity, etc.

Starting from the (revised) title, I am troubled by the use of elasticity. As these authors likely know, elasticity is a term that already has a well-established mathematical definition. It would be one thing if they were to acknowledge this in their paper and, upon first use, provide a clear definition of their own use of the term. But they fail to do this, to the detriment of their study. If I understand things correctly, what the authors wish to imply when they say elasticity is (a) that the system moves away rapidly from a previous "resting state" upon perturbation and yet (b) that system returns to the resting state similarly rapidly---at least on a macroscopic scale. If we are to stick to the ecological analogy, both (a) and (b) above already have terms in the ecological

literature which are widely used and widely accepted: reactivity (Neubert & Caswell 1997 Ecology) and resilience (Holling 1973 Annual Rev Eco Syst). Though I do not object to the authors using "elastic" to refer to a system that is both reactive and resilient, the fact that they do not echoes Reviewer #4's concern that the "analogies used at the best inaccurately [describe]" what they measure.

2) Communication networks as mutualistic assemblages

Upon first glance, I am strongly inclined to agree with Reviewer #4 that this system is more akin to consumer-resource dynamics than to mutualistic assemblages. Or maybe there is simply a symbiotic relationship and we need provide an a priori definition of the sign of the interaction. Reviewer #4 in particular focused on the passive nature of memes as the basis for their argument against mutualism; this is additionally supported by the authors' modelling within which users rewire their interactions to maximise their abundance. Indeed, users can choose memes to propagate or create, but memes clearly cannot choose users. On the other hand, from this perspective I can see how memes might well "benefit" from being used more frequently, and how they might "compete". The same is certainly not true for resources in the classical consumer-resource context. In contrast, I personally found the statement that users in the real world select memes specifically to increase their "abundance" far more tenuous or specious. The authors do not appear to provide any references to back this up nor do they have any data with which to back up this claim. Perhaps retweets or likes could serve the purpose of quantifying the "benefit" to users, but these are not studied here. In that vein, the closest, or most parsimonious, match may well be commensalism.

Of course, worrying about how any potential benefit could be measured probably detracts from what I feel is a much more salient point: in the introduction, the authors present and argue the analogy to mutualistic assemblages as if it is incontrovertible fact. And yet absolutely nothing in their paper demands that this be the case. The paper would be drastically improved, for example, if they rearranged the introduction such that the analogy and modelling are, at best, presented as conjectures about the how and why of the emergent patterns they find in the empirical data. In a sense, they pre-empt the possibility that a mutualistic model could capture the dynamics---which is a result and pertinent to the discussion---by putting it in the introduction when it is still remains highly speculative. In reality, much of the paper suffers from similar issues where even the most minor observations are "remarkable" or where what frequently is little more than qualitative, visual comparison between a model and data "[proves] that an ecological framework...is a powerful tool" or "[brings] to light the precise mechanisms causing the observed topological reorganisation". Truth be told, the authors cannot prove anything with their analysis, nor can they state that they have uncovered a causal mechanism. Their model helps them postulate a possible generative mechanism that produces qualitatively similar patterns. This is nothing to be scoffed at, but is also not causal. If the authors could resist the urge to oversell their work over and over again, the manuscript would better stand on its true merits---and allow those merits to shine for what they are---as opposed to watering things down with one grandiose claim after another.

3) Ecosystems and coevolution

As previously noted, the authors appear both extremely open-minded in terms of obtaining inspiration from other fields while also surprisingly resistant to suggestions that might make the mapping between their work and that of others crisper than it is. On the one hand, they rightfully follow the advice that there is no such thing as a mutualistic ecosystem: there are mutualistic assemblages (i.e., collections of agents that interact in a mutualistic fashion) and mutualistic networks (i.e., the mathematical objects often used to describe the interactions that occur within mutualistic assemblages). On the other hand, they argue that their use of coevolved and/or coevolution is correct despite Reviewer #4 providing a clear explanation for why this is inappropriate. Moreover, the best evidence they could find to support their cause is drawn from publications in physics and not ecological/evolutionary journals. As with the above, I cannot see

what is to be gained from digging in; there is absolutely no harm whatsoever in changing these terms to "interplay of structure and dynamics in networks", "feedback between structure and dynamics in networks", or to simply refer to "adaptive dynamics". Not only will this be more accurate, but it will avoid the risk of alienating a large section of the presumed target audience for this paper: biologists.

Minor comments

1) Nature Communications is a broad, multidisciplinary journal and hence articles published there should be as accessible as possible. At present, I do not feel that the manuscript has been written or revised to be as accessible as possible. Possible "abuses" of language aside, the grammar appears correct, but the text is also extremely prolix. The first sentence of the Discussion is but one of many examples where an equivalent message could almost certainly be conveyed with far fewer words.

2) In their response to Reviewer #2's previous comment 2, the authors argue against studying other indicators beyond nestedness and modularity. In particular, I'm surprised that they wouldn't also include plots of meme/hashtag evenness (or Shannon diversity) as a function of time. They are correct that a heterogeneous distribution of abundance (or degree) is not a sufficient condition for nestedness, but it is a sufficient one and hence may be as useful as Q or N (Bluthgen 2008 Ecology). Even though their networks are qualitative, they do still have data for meme usage and hence should be able to measure and present this pattern. It's also been known for some time that networks may appear qualitatively nested while also being qualitatively anti-nested (Staniczenko 2013 Nature Comms)

3) When Reviewer #4 pointed out that within-module nestedness had first been pointed out by Lewinsohn over a decade ago, I suspect part of their motivation was that that study and other precursors be cited appropriately in the manuscript to give credit where credit is due. That is, the authors currently only cite Lewinsohn's study and one of their own papers when introducing the topic, potentially giving the false impression that nothing else had been published on the topic in the intervening years. In addition to the new citations (which only appear in the Methods), I would also encourage the authors to at least cite Kondoh et al. Ecology (2010) who quantified this pattern in non-mutualistic settings.

4) The authors interchange between actors and users, memes and hashtags, at seemingly random moments throughout the text. It would seem to be helpful to pick one nomenclature and stick to it.

5) Were the 2000 most active users the same 2000 users for each of the temporal slice that created a bipartite network? As far as I could tell, this was never specified, but it would seem to be very important for this to be the case. Otherwise, it may not be the same users reorganizing/rewiring and hence does not map nearly as well to the authors' simulations.

6) Along similar lines, Table S1 gives an indication that the top 2000 users is only a tiny subset of the total user-hashtag assemblage covered by the authors' dataset. How sensitive are their results to this threshold and on what basis was this chosen as the cutoff?

7) In order to clearly outline why the mutualistic population-dynamics model is a fair representation of the system, it would help to have some rationale provided for why users and memes might directly compete with each other as captured by the beta coefficients in their model. In a plant-pollinator network, for example, this is included because plants compete for resources (e.g., light, nutrients) beyond pollinators, and because pollinators compete for resources (e.g., other food sources, habitat) beyond pollen. What is the equivalent here?

8) What is actually on the y-axes of Figure 4b and 4d? It clearly cannot just be frequency as

labelled because you cannot have a negative frequency. Is this $\log(\text{frequency})$?

9) The authors generate synthetic data with average connectance 10^{-2} . Why?

10) The authors' analyses focus on externally defined "shocks" to the assemblage, but there are many other windows within which the networks appear to experience shifts from predominantly modular to predominantly nested and back. In Figure 1a, for example, this appears around 04/16 or right after the first shock they consider. If structural elasticity is only observable after the fact when t_E has been defined, doesn't the utility of the authors' findings diminish considerably? If not, couldn't the size of endogenous fluctuations in these measures at least be used to estimate whether or not the responses to perturbations are statistically remarkable/unexpected?

Nature Communications - response to reviewers (II)

Manuscript Number: NCOMMS-20-18890A

Structural Elasticity in Online Communication Networks: an ecological approach

REVIEWER COMMENTS

Reviewer #1:

In the revised version of the manuscript the authors could clarify basically all my concerns with an impressive amount of extra analysis. Even though most links to the data remain qualitative, all the comparisons are reassuring that the empirical observations can be linked to the proposed mechanisms, which seem to describe an interesting and potentially quite general behaviour during shock events on social media.

In particular I am convinced that merely an increase in activity can result in the observations, as the ensemble simulations in the response have shown.

I am convinced that this work presents a robust interpretation, through an innovative model, for an interesting qualitatively similar response of the social media ecosystem across different topics.

We thank the referee for the very positive assessment of our work. We are glad to see that our efforts to respond to their insightful questions have rendered a stronger work, and we acknowledge and thank their role in pursuing more robust and convincing results.

Reviewer #3:

The authors have addressed my comments in a successful way. I am satisfied with their revision in terms of social sciences. Congratulations!

We thank the referee for their words, and for recommending the publication of the manuscript in Nature Communications.

Reviewer #5:

I have read the manuscript by Palazzi and coauthors with considerable interest. I think that their analyses are quite novel and potentially very interesting, and on the whole appear to be properly conducted.

We thank the reviewer for the positive comments on the value of the manuscript. Below we offer a point-by-point response to address the issues raised by the referee.

This notwithstanding, I am in very strong agreement with Reviewers 2 & 4 from the previous round of review that the links to the ecological literature need considerable improvement before I could regard the manuscript as publishable. In many ways, I feel that this is a direct consequence of the manner in which the authors have decided to present the study; that is, if they have found themselves between a rock and a hard place after peer review, the current presentation of the study is as much to blame as any use or abuse of language.

I agree with the authors that exchange between disciplines can lead to very fruitful outcomes and hence cross pollination should not be discouraged (all puns intended). However, it is important to get things right (or as right as possible) so that subsequent researchers have firm ground to stand on, and to avoid reinventing the wheel. Since these authors emphasise "an ecological approach" in their title, it behoves them to get the language right---not from their own perspective but from the perspective of domain experts. Right now, I do not feel their manuscript accomplishes this feat, and the apparent resistance to change (adaptation?) that comes across in their response to reviewers does them few favours. Assuming that everyone can get on the same page in terms of the underlying language (see below), I also have some specific suggestions that might at least "soften" the analogy to ecology to the point that it becomes acceptable (or at least tolerable).

We apologise if our previous responses could be interpreted as resistance to change. Rather, those responses possibly reflect a "tension" that is inherent to the work: the attempt to harmonise different scientific languages --Ecology, Complex Systems and, to a lesser extent, Social Science.

1) Resilience, elasticity, structural elasticity, etc.

Starting from the (revised) title, I am troubled by the use of elasticity. As these authors likely know, elasticity is a term that already has a well-established mathematical definition. It would be one thing if they were to acknowledge this in their paper and, upon first use, provide a clear definition of their own use of the term. But they fail to do this, to the detriment of their study. If I understand things correctly, what the authors wish to imply when they say elasticity is (a) that the system moves away rapidly from a previous "resting state" upon perturbation and yet (b) that system returns to the resting state similarly rapidly---at least on a macroscopic scale. If we are to stick to the ecological analogy, both (a) and (b) above already have terms in the

ecological literature which are widely used and widely accepted: reactivity (Neubert & Caswell 1997 Ecology) and resilience (Holling 1973 Annual Rev Eco Syst). Though I do not object to the authors using "elastic" to refer to a system that is both reactive and resilient, the fact that they do not echoes Reviewer #4's concern that the "analogies used at the best inaccurately [describe]" what they measure.

We thank the referee for the comment, which makes it clear that our choice of 'elasticity' is troublesome (as it happens, the term 'elasticity' was also discussed in the first reviewing round: referee 3 understood the term as it is used in Economy, while we clearly had another concept in mind).

We find the idea of using 'reactivity' a very appealing one. Its attention to the short-term, transient behaviour of the system following a perturbation captures well our aim to characterise rapid changes in the structure of online communication systems. However, the concept stems directly from ecological stability theory, which deals with the system's dynamics near an equilibrium. Instead, the term we are looking for is to be applied on the structure on the network. Also, our results in section "Lasting effects of perturbations on the system's dynamics" suggest that the capacity of the system to stick to certain structural arrangements might not be mirrored at the dynamical level. Then, the use of 'reactivity' might be misleading (and the same applies to 'resilience').

Triggered by the referee's comment, we have searched for an appropriate term in the Ecological literature, avoiding 'elasticity', 'reactivity' or 'resilience' for the above reasons. The text finally sticks to the word 'flexibility', from the very recent work (now cited as ref. [33]):

CaraDonna, P. J. & Waser, N. M. Temporal flexibility in the structure of plant–pollinator interaction networks. *Oikos* <https://doi.org/10.1111/oik.07526> (2020).

Here, with the word 'flexibility' the authors explicitly refer to short-term variations in network structural properties.

2) Communication networks as mutualistic assemblages

2.1. Upon first glance, I am strongly inclined to agree with Reviewer #4 that this system is more akin to consumer-resource dynamics than to mutualistic assemblages. Or maybe there is simply a symbiotic relationship and we need to provide an a priori definition of the sign of the interaction. Reviewer #4 in particular focused on the passive nature of memes as the basis for their argument against mutualism; this is additionally supported by the authors' modelling within which users rewire their interactions to maximise their abundance. Indeed, users can choose memes to propagate or create, but memes clearly cannot choose users. On the other hand, from this perspective I can see how memes might well "benefit" from being used more frequently, and how they might "compete". The same is certainly not true for resources in the classical consumer-resource context. In contrast, I personally found the statement that users in

the real world select memes specifically to increase their "abundance" far more tenuous or specious. The authors do not appear to provide any references to back this up nor do they have any data with which to back up this claim. Perhaps retweets or likes could serve the purpose of quantifying the "benefit" to users, but these are not studied here. In that vein, the closest, or most parsimonious, match may well be commensalism.

We appreciate these thoughtful reflections on the nature of the interactions between actors and memes. We now see that the three relationships (meme-meme, actor-actor, and actor-meme) have not been sufficiently justified.

1) Competition among memes

The idea that words compete to be used by speakers is one of the fundamentals of cultural evolution, a dominant framework to explain language change in the last three decades. Although the accounts within this discipline differ widely in their theoretical background and assumptions, they share the postulate that linguistic units are 'replicators': their survival depends on their ability to be copied, i.e., internalised by other speakers of the language. As it happens, Darwin himself subscribed to this view in *Descent of Man* (1871), when he wrote that "the survival or preservation of certain favoured words in the struggle for existence is natural selection".

Competition presupposes of course some resource scarcity, and lexical theorists argue that the resource in question is the speakers' attention (users, in the Twitter context). Often this is just implicit, or else they refer generically to cognitive resources. Words, in this context, evolve to become compressible (reduction or simplification) and discriminative (maintaining relevant distinctions). In either case, the underlying assumption is that the cognitive system "wishes for" easier-to-internalise terms.

We have added a few sentences on these ideas, along with the following references refs. [14-17] in the new version:

McMahon. *Understanding Language Change*. Cambridge: Cambridge University Press. (1994).

Brighton, Henry, Kenny Smith and Simon Kirby. Language as an evolutionary system. *Physics of Life Reviews*, 2(3), 177–226 (2005).

Zaslavsky, Noga, et al. Efficient compression in color naming and its evolution. *Proceedings of the National Academy of Sciences* 115.31 (2018): 7937-7942.

Monica Tamariz. Experimental Studies on the Cultural Evolution of Language. *Annual Review of Linguistics*, 3 389-407 (2017).

All of the previous is not meant to explain lexical competition in online environments. We believe however that the logic underlying these developments hold. Still, to strengthen the idea that memes compete for attention also online, we have added a comment and a reference (Lorenz-Spreen *et al.* Nature Communications 2019) (ref. [1]). This work was already cited in a different context, but not specifically as part of the argument on meme competition.

2) Competition among actors

On the other hand, it is true that we didn't provide evidence that users select contents specifically to increase their visibility (or abundance), and this had to be corrected. We now resort to the literature to back the idea that users make decisions on what digital contents to share based, at least partially, on the expected visibility they will obtain. These new references point at the motivational and neurological reasons of such behaviour. On the motivational side, we include the following work (ref. [20] in the new version):

Malik, Aqdas, Amandeep Dhir, and Marko Nieminen. "Uses and gratifications of digital photo sharing on Facebook." *Telematics and Informatics* 33.1 (2016): 129-138.

Mostly based on surveys, such literature emphasises that participation in the social network (Twitter, Flickr, Facebook, etc.) is not so much about having general discussions with others, but as seeking attention through content distribution. All in all, the importance of activity seems to be that it enables the promotion of oneself.

In a different fashion, neurological research provides quantitative evidence of the effects of varying peer endorsement (tantamount to visibility) on social media. Such research pays attention to the patterns that emerge in the brain's reward system. When engaged in social activity, individuals pursue socially valued or rewarding outcomes --approval, acceptance, reciprocity, recognition-- as a means toward learning about others and fulfilling social needs of forming meaningful relationships. Notably, these values (approval, acceptance, reciprocity, recognition) are aspects of (or presuppose) visibility. In more practical terms, a retweet (or a 'like' on Facebook, etc.) shares features with both monetary and social rewards as a means of feedback that shapes reinforcement learning. Thus we now also cite (refs. [19, 21] in the new version):

Fareri, Dominic S., and Mauricio R. Delgado. "Social rewards and social networks in the human brain." *The Neuroscientist* 20.4 (2014): 387-402.

Sherman, Lauren E., et al. "What the brain 'Likes': neural correlates of providing feedback on social media." *Social cognitive and affective neuroscience* 13.7 (2018): 699-707.

3) Mutualistic actor-meme interactions

The last type of relationship that we assume in our work is admittedly the most controversial and speculative --the one our work aims to unveil. To us, the existence of mutualistic links is implied

in the existence of the competitive ones: in a communication environment, the best tool for an actor to compete with its peers is precisely to build the best possible discourse; and the best way for a meme to compete with other lexical candidates is to be clear and concise: thus the mutualistic relationship.

Note that the use of 'better' memes does not discard other strategies to maximise visibility, but these are not considered in our framework: engaging with other users' content (seeking reciprocity), URL sharing, image and video posting, etc.

2.2. Of course, worrying about how any potential benefit could be measured probably detracts from what I feel is a much more salient point: in the introduction, the authors present and argue the analogy to mutualistic assemblages as if it is an incontrovertible fact. And yet absolutely nothing in their paper demands that this be the case. The paper would be drastically improved, for example, if they rearranged the introduction such that the analogy and modelling are, at best, presented as conjectures about the how and why of the emergent patterns they find in the empirical data. In a sense, they pre-empt the possibility that a mutualistic model could capture the dynamics---which is a result and pertinent to the discussion---by putting it in the introduction when it still remains highly speculative. In reality, much of the paper suffers from similar issues where even the most minor observations are "remarkable" or where what frequently is little more than qualitative, visual comparison between a model and data "[proves] that an ecological framework...is a powerful tool" or "[brings] to light the precise mechanisms causing the observed topological reorganisation". Truth be told, the authors cannot prove anything with their analysis, nor can they state that they have uncovered a causal mechanism. Their model helps them postulate a possible generative mechanism that produces qualitatively similar patterns. This is nothing to be scoffed at, but is also not causal. If the authors could resist the urge to oversell their work over and over again, the manuscript would better stand on its true merits---and allow those merits to shine for what they are---as opposed to watering things down with one grandiose claim after another.

At the time of writing the introduction, our intention was to follow precisely the path that the referee points at with their comment. The skeleton of our argument could be phrased following the "classical" scientific cycle: observation, hypothesis, model, prediction. From the referee's comment, we see that we have failed at delivering this argumentative path, rather presenting as a given fact that online social media and mutualistic assemblages operate in the same way.

Thus we have rewritten part of the introduction, to convey the idea that the work begins with an observation (empirical emergent patterns of structural flexibility in Twitter datasets), which leads to a hypothesis (competitive and mutualistic interactions, together with visibility maximisation, govern the system) that we encapsulate in a modelling framework. Finally, we show that the model indeed reproduces the most salient features of the initial observations, and even delivers a prediction that the data do not falsify.

3) Ecosystems and coevolution

As previously noted, the authors appear both extremely open-minded in terms of obtaining inspiration from other fields while also surprisingly resistant to suggestions that might make the mapping between their work and that of others crisper than it is. On the one hand, they rightfully follow the advice that there is no such thing as a mutualistic ecosystem: there are mutualistic assemblages (i.e., collections of agents that interact in a mutualistic fashion) and mutualistic networks (i.e., the mathematical objects often used to describe the interactions that occur within mutualistic assemblages). On the other hand, they argue that their use of coevolved and/or coevolution is correct despite Reviewer #4 providing a clear explanation for why this is inappropriate. Moreover, the best evidence they could find to support their cause is drawn from publications in physics and not ecological/evolutionary journals. As with the above, I cannot see what is to be gained from digging in; there is absolutely no harm whatsoever in changing these terms to "interplay of structure and dynamics in networks", "feedback between structure and dynamics in networks", or to simply refer to "adaptive dynamics". Not only will this be more accurate, but it will avoid the risk of alienating a large section of the presumed target audience for this paper: biologists.

We can only agree with the reviewer. Our zeal to grant the presence of certain expressions harmed, instead of strengthened, the objectives of the work. We have followed the referee's advice and substituted the mentioned expressions as advised.

Minor comments

1) Nature Communications is a broad, multidisciplinary journal and hence articles published there should be as accessible as possible. At present, I do not feel that the manuscript has been written or revised to be as accessible as possible. Possible "abuses" of language aside, the grammar appears correct, but the text is also extremely prolix. The first sentence of the Discussion is but one of many examples where an equivalent message could almost certainly be conveyed with far fewer words.

We have tried to simplify parts of the text, employing a more direct language or removing superfluous text.

2) In their response to Reviewer #2's previous comment 2, the authors argue against studying other indicators beyond nestedness and modularity. In particular, I'm surprised that they wouldn't also include plots of meme/hashtag evenness (or Shannon diversity) as a function of time. They are correct that a heterogeneous distribution of abundance (or degree) is not a sufficient condition for nestedness, but it is a sufficient [necessary] one and hence may be as useful as Q or N (Bluthgen 2008 Ecology). Even though their networks are qualitative, they do still have data for meme usage and hence should be able to measure and present this pattern. It's also been known for some time that networks may appear qualitatively nested while also being qualitatively [quantitatively] anti-nested (Staniczenko 2013 Nature Comms)

On reading this comment, we now understand better the motivations of referee #2 when suggesting the use of alternative measures, and Shannon entropy in particular. If only for this, we thank the reviewer for the additional context. In particular, we realise now that the question is not just about observing a nested pattern in the binary matrix, but actually to take into account the weights in it --something that certainly we had overlooked.

The top panels in Figure R1 below show two (left and center) user-hashtag matrices, which correspond to the Spanish dataset in two different moments. In one of them, the predominant pattern is in-block nested (left); in the other (center), the predominant pattern is purely nested. The matrices have been arranged to facilitate the visualisation of the corresponding architecture, and the cells are coloured according to the link weight (i.e. how many times user u tweeted hashtag h). Following the arguments in Staniczenko (Nat Comm, 2013), these matrices provide a visual insight on whether the qualitatively observed nested patterns are also quantitatively matched. Indeed, one can observe that the strongest links (darker colours) correspond to those among generalists. To better appreciate this pattern in the case of global nestedness, we draw on the right a submatrix for the first 150 users (and corresponding hashtags). Note that Fig. R1 is now included as Figure S5 in the Supplementary Materials (along with a new section 1.5 in the same document).

Figure R1: Top panel: weighted in-block nested (left) and nested (center) matrices, in which darker colours correspond to stronger links. The right-most matrix corresponds to the subset of most generalist users/hashtags in the center, to highlight the stronger weights in the upper-left part. If only visually, it is apparent that the matrices are qualitatively and quantitatively in-block nested and nested, respectively. Lower panel: evolution of evenness for the Spanish dataset. As expected, evenness remains relatively high (baseline behaviour), with abrupt dips whenever external shocks affect the system.

The bottom panel shows the evolution of evenness as defined in

Blüthgen, Nico, et al. "What do interaction network metrics tell us about specialization and biological traits." *Ecology* 89.12 (2008): 3387-3399.

which is now included as ref. [54] in the manuscript, along with ref. [55] to Staniczenko's work as well. As expected, evenness portrays a similar evolution to those of nestedness and modularity: it remains relatively stable in all the period, except for exceptional episodes that disrupt the attention focus in the system.

We believe that this is a truly interesting aspect. In particular, we suspect that evenness might be an interesting measure to discriminate between nested, in-block nested and modular-only arrangements. This, of course, demands a dedicated effort and an appropriate benchmark (i.e. toy model), which lies beyond the scope of the work.

3) When Reviewer #4 pointed out that within-module nestedness had first been pointed out by Lewinsohn over a decade ago, I suspect part of their motivation was that that study and other precursors be cited appropriately in the manuscript to give credit where credit is due. That is, the authors currently only cite Lewinsohn's study and one of their own papers when introducing the topic, potentially giving the false impression that nothing else had been published on the topic in the intervening years. In addition to the new citations (which only appear in the Methods), I would also encourage the authors to at least cite Kondoh et al. Ecology (2010) who quantified this pattern in non-mutualistic settings.

We have reviewed the locations of the mentioned citations, not only the one from Lewinsohn, but also those that have addressed the problem of hybrid structures in the past. Due to relocation, these works are now cited as refs. [40-46] in the Introduction. We note that, in our first revision of the manuscript, even more references on the topic were introduced. Also, we thank the referee for suggesting this new reference, which we were unaware of and now include as ref. [41].

4) The authors interchange between actors and users, memes and hashtags, at seemingly random moments throughout the text. It would seem to be helpful to pick one nomenclature and stick to it.

This is a timely advice that has also been a matter of debate among ourselves. The debate stems from the fact that the model is not meant to be a “Twitter model”, but rather a more general framework for similar online social platforms: thus, a more general nomenclature is needed (actors, memes), as *user* and *hashtag* are rather platform-specific. On the other hand, all the empirical evidence that we offer is from Twitter, and we should stick to this limitation.

Considering this, we have limited our use of ‘actors’ and ‘memes’ to the Introduction and Discussion (where the discourse is on general communicative contexts), and ‘users’ and ‘hashtags’ in the rest of the paper, when the focus is set on the results over Twitter datasets.

5)

5.1. Were the 2000 most active users the same 2000 users for each of the temporal slice that created a bipartite network? As far as I could tell, this was never specified, but it would seem to be very important for this to be the case.

This is certainly a delicate aspect of the work that demands the maximum clarity. In our previous version, we had two references regarding the continuity of nodes (not only users, but also hashtags) across slices. The first one in the main text (Methods summary):

*“Then, for the empirical case, for each dataset, we split the Twitter stream into chunks according to non-overlapping time windows with three hours of duration $\omega=3h$, containing the 2000 most active unique users, **while the number of hashtags is variable (depending on the amount produced by those 2000 users).**”*

It is implicit in the highlighted sentence that hashtags may change from one snapshot to the next. Even more: since hashtags are not chosen, but strictly determined by what users tweeted on, the overlap between consecutive time windows could be (but never is) 0.

Then, in the SM (section 1.2), the text is more explicit:

“It is also important to highlight that the $A^{\{t\}}_{uh}$ matrices may not contain the same nodes across t : as time advances, users join (disappear) as they start (cease) to show activity; the same applies for hashtags, which might or might not be in the focus of attention of users.”

This information, admittedly, should be clear also in the main text (we have added a clarification in the Methods section).

For the sake of completeness, let us quantify the temporal continuity of users and hashtags over time. On the hashtags side, Figure R2 (top) represents the overlap between consecutive hashtag coversets ($|H_t \cap H_{t-1}| / \min(|H_t|, |H_{t-1}|)$), showing a strong persistence of the main hashtags in general, and above 60% during the highlighted events (the trend is smoothed to remove the fluctuations during the night periods, in which activity remains very low; the raw plot reaches a maximum at ~80%, and a minimum at ~20%).

Figure R2: Overlap across time between the most relevant hashtags (top) and users (bottom) in consecutive snapshots of the Spanish dataset.

Figure R2 (bottom) shows, conversely, the overlap between consecutive user coversets (that is, $|U_t \cap U_{t-1}| / \min(|U_t|, |U_{t-1}|)$), which remains relatively constant and low between 10 and 15% (again, the trend is smoothed). This volatile situation (both for users and hashtags) is quite normal in time-resolved ecology field studies, where the emphasis is placed on the system's dynamics rather than individual species. In our case, despite the composition of users, hashtags, and their interactions may vary across minutes, hours, and days, evidence suggests that networks are composed of a reliable core of generalist 'species', accompanied by a changing suite of specialist 'species'. Along the same lines in Ecology, see for instance (new ref. [23] in the SM):

Zografou, Konstantina, et al. "Stable generalist species anchor a dynamic pollination network". *Ecosphere* 11.8 (2020): e03225.

Figure R2 and surrounding text are now included in Section 1.2 of the Supplementary Materials.

5.2. Otherwise, it may not be the same users reorganizing/rewiring and hence does not map nearly as well to the authors' simulations.

The referee raises a legitimate doubt about the suitability of the modelling approach: the problem is not so much that the nodes change in time (comment 5.1 above), but the failure of the model to include this feature in particular. We respectfully disagree with the referee, and we provide some evidences against their argument.

The first fact that we need to take into account is that, in our synthetic simulations, we always have a privileged global view of the system, in which nodes cannot enter or leave the observed stage. That is, Q , \mathcal{N} and \mathcal{I} are measured including all the nodes and their interactions. For a fair comparison with the data in this aspect, we should consider a fraction of the synthetic system, and then see whether the top users (or hashtags) do change it time or not.

This is what Figure R3 below precisely shows. To obtain it, we take the whole history of the synthetic simulation shown in Figure 3 of the main text. Keep in mind that the system is made up of $N_U = N_H = 100$. Here, as we do with empirical data, we assume that we can only manage a partial observation of the system, so for each temporal slice we select the 50 most active (abundant) users, and the corresponding hashtags that they cite (which may be any number from 0 to 100). In the top and central panels, we show the overlap between slices for hashtags (top) and users (central). We can see that these overlaps are, in general, smaller than 1; and these changes are even more drastic after the introduction of an external shock ($t = 3 \times 10^4$), when the overlaps hit the minimum.

Figure R3: Overlap across time between the hashtags (top) and users (center) in consecutive snapshots, when only the top 50% of the users are taken into account. Bottom: the partial view of the system does not blur the emergent structural patterns reported in Figure 3 of the main text.

The fact that the overlap in this experiment is significantly larger than the empirical one, i.e. Figure R2, is an effect of the small size of the synthetic experiment (both in terms of nodes, and number of topics). We remark that here we are systematically considering half of the users (and often more than half of the hashtags), which is not the case in general when analysing empirical datasets (see Figure R4 below).

Finally, we highlight that the partial pruning of the system does not affect the observation of emergent structural patterns: the bottom panel of Fig. R3 displays the evolution of Q and \mathcal{N} , which closely follow the trends in Figure 3 of the main text.

Summarising, the apparent failure of the model to match the observed species turnover is a consequence of the limited computational capabilities: synthetic simulations, unlike real

datasets, always permit a privileged global view of the system, in which nodes cannot enter or leave the observed stage. That said, we insist that the model needs, in future efforts, to include birth/death/invasion processes, because species turnover in empirical data is not just a matter of network pruning (i.e. which nodes are included in the network), but also of nodes that join and leave the system (if only because of circadian rhythms).

This discussion has been included in the Supplementary Materials (new Section 3.5).

6) Along similar lines, Table S1 gives an indication that the top 2000 users is only a tiny subset of the total user-hashtag assemblage covered by the authors' dataset. How sensitive are their results to this threshold and on what basis was this chosen as the cutoff?

The referee tackles an important issue here, which connects also with the previous comment 5.2. The large size of our several datasets handicaps the data processing and makes the calculations time-consuming. We must therefore apply some restrictions to the number of users (N_U) considered in the network.

Taking the comment literally ("*Table S1 gives [...] covered by the authors' dataset*"), it is manifestly true: in the best case (Catalan dataset), $N_U = 2000$ users represent 2.5% with respect to an aggregate of all users over that dataset; in the worst (Charlie Hebdo), the same amount of users hardly reach 0.1% of the total.

However, we infer that the comment is related not to the aggregate user count, but to each snapshot that the paper analyses, i.e. 3-hours time windows (15-minutes windows during the selected events). In this case, we argue that $N_U = 2000$ users (and the number of hashtags produced by those, which varies from slice to slice) are a good representative of the overall activity and of the structural patterns that emerge. At this point, it is worth noting that, by selecting top contributors (and their associated hashtags), we guarantee that both generalists and specialists will show up –if any nested pattern is to be found. Also, the chances of obtaining a connected matrix are higher (note that the appearance of disconnected components would render a trivially modular network). In the left panel of Figure R4 below, we show, for the Spanish dataset, the corresponding proportion of the 2000 unique users (ρ), with respect to the total number of users for each snapshot. That is: if the 3-hour window at time t has N_t users, the panel is showing N_U / N_t . Clearly, 2000 users represent in general a minority of all the participants –with an average of ~40%. In the right panel, however, we show the proportion of tweets for which those 2000 users are responsible (Tw_U / Tw_t). It turns out that, for each 3-hour window, these top contributors often account for more than 60% of the overall activity, with an average around 57%. Note that, to avoid the heavy circadian fluctuations (at night, $N_U = 2000$ users represent 100% of the total), both panels are shown as the average on a sliding window scheme.

Figure R4: Fraction of users N_U / N_t (where N_t is the total number of users in a window at time t), which shows that $N_U = 2000$ represents, most of the time, a minority of the total (left). However, their status as top contributors grants that most of the activity for a given window is captured by that minority (right).

On the other hand, one might also wonder to what extent the results we present are robust against other network sizes. Although we did not report on them, we have experimented with different thresholds, i.e. $N_U = \{500, 1000\}$ in the case of the Spanish dataset. For these numbers of users, the evolution of Q and \mathcal{N} was generally indistinguishable from the one reported in Fig. 1 of the main text, see Figure R4 below for $N_U = 500$.

Figure R4: Nestedness and modularity evolution for the Spanish dataset, where each 3-hour snapshot is built enforcing a threshold of $N_U = 500$ users. As in Figure 1 of the main text, the values for Q and \mathcal{N} are shifted to the initial one.

We now comment on these results in the Supplementary Materials, Section 1.6 and Figures S6 and S7, respectively.

7) In order to clearly outline why the mutualistic population-dynamics model is a fair representation of the system, it would help to have some rationale provided for why users and memes might directly compete with each other as captured by the beta coefficients in their model. In a plant-pollinator network, for example, this is included because plants compete for resources (e.g., light, nutrients) beyond pollinators, and because pollinators compete for resources (e.g., other food sources, habitat) beyond pollen. What is the equivalent here?

This is a very interesting question which we had not treated in depth before.

Strictly speaking, users do not compete for memes. Unlike pollen, memes *are* an infinite resource in the sense that nothing prevents the usage of a meme, regardless of the amount of times it has been used before. On the other hand, as stated in comment 2.1 above, the visibility that users compete for actually encapsulates many other aspects (recognition, approval, etc.). In more quantifiable terms, these aspects may include retweets, mentions, likes, increasing the number of followers, or increasing the number of lists in which a user is included (what some scholars tag as ‘social media capital’). Thus, aligned with the referee’s idea, visibility aggregates many dimensions of competitive behaviour.

Beyond general user-user competition, our model incorporates a λ parameter which tunes the amount of intra-topic competition. Translated to the analogy above, there is indeed a ‘habitat’ competition when two users delivering messages on the same topic aim for higher abundances.

On the other hand, competition among memes can take many forms (see response to comment 2.1) --but ultimately there is only one finite resource through which they can thrive: a cognitive system (discarding the presence of bots). Therefore the analogy of light or nutrients cannot be mapped to the online communication context.

These are yet further evidence that a one-to-one mapping is not possible between natural mutualistic assemblages and online communication systems --an idea that we try to convey throughout the work.

8) What is actually on the y-axes of Figure 4b and 4d? It clearly cannot just be frequency as labelled because you cannot have a negative frequency. Is this $\log(\text{frequency})$?

We thank the reviewer for this observation, and we apologise since, in fact, we have failed to clarify this aspect. For better visualization of the reported behaviour, the frequency in these plots was shifted with respect to the initial one (i.e., the observed frequency at the beginning of the plot). We have added a clarification in the caption of the affected panels.

9) *The authors generate synthetic data with average connectance 10^{-2} . Why?*

In Sec. 2.7 and Fig S5 of the Supp Information we explain:

“To avoid excessive computational costs, we consider small synthetic networks of $N_U = 100$ users and $N_H = 100$ hashtags with random connections across guilds, and density (connectance) $C \sim 10^{-2}$. We do so to match the same order of magnitude of empirical networks when we take $N_U = 100$, see blue triangles in Fig. S5.”

We now see that this was only clarified in the Supplementary Materials, so we have now added this information to the main text (Methods section).

10) *The authors' analyses focus on externally defined "shocks" to the assemblage, but there are many other windows within which the networks appear to experience shifts from predominantly modular to predominantly nested and back. In Figure 1a, for example, this appears around 04/16 or right after the first shock they consider. If structural elasticity is only observable after the fact when t_E has been defined, doesn't the utility of the authors' findings diminish considerably? If not, couldn't the size of endogenous fluctuations in these measures at least be used to estimate whether or not the responses to perturbations are statistically remarkable/unexpected.*

This is an interesting observation that we failed to comment in the previous versions of the manuscript. Surely, when monitoring the evolution of empirical data, transitions to/from nested/modular arrangements occur more often than the few ones for which we provide an explanation (political debate, election day, etc.).

We humbly claim that this does not diminish, but rather increases, the utility of the work: precisely the existence of those unexplained transitions signal an attention shift that has occurred in the online discussion --independently of us being able to find an echo of such events in the media. Following the example raised by the referee: surely, there is a large shock on the day after the electoral debate, which (in terms of nestedness increase) is even larger than the one on the day that the polling took place. Some mining in the Spanish newspapers around that date tells us that, following the debate on April 22 on the public Spanish TV (RTVE), there was a second debate carried out by a subset of the candidates in Atresmedia (private TV company, broadcasted through Antena 3 and La Sexta channels) the very next day. See the related news:

<https://www.rtve.es/noticias/20190424/debate-electoral-generales-directo-ultima-hora/1927280.shtml>

<https://www.lavanguardia.com/politica/20190422/461778531739/horario-debate-atresmedia-elecciones-generales-sanchez-iglesias-rivera-casado.html>

Even the minor nestedness peak on April 16-17 can be traced to identify an underlying discussion: on that evening the dates of both debates were announced, and also the fact that the second debate would exclude certain candidates (so that only the most representative parties, in terms of congress seats, were represented) was known. Echoes of that polemic decision could be read in the online newspapers updates:

<https://www.elmundo.es/espana/2019/04/16/5cb6158021efa01a5d8b4593.html>

Similarly for other datasets. For example, on the Nepali dataset, we chose to develop our arguments around the largest earthquake aftershock (May 12th), but other events seemingly affected the online discussion before and after that: several minor aftershocks, news on the international humanitarian response, etc.

Thus, from a practical perspective, the detection of structural changes is tantamount to the detection of collective attention foci (or lack thereof). And indeed, as the referee points out, a proper statistical analysis of these fluctuations could serve the purpose of distinguishing true shifts from mere fluctuations in the system (along these lines, we refer to section 3.2 of the Supplementary Materials, where the statistical significance of Q and \mathcal{N} is discussed).

We have included a brief paragraph in the results section, and in the discussion, regarding the ability of the system to detect these collective attention episodes, which can be helpful to track the state of the system (with or without the traditional media reflecting it).

Reviewers' Comments:

Reviewer #5:

Remarks to the Author:

I feel that the revised version of the manuscript has clarified the vast majority of my concerns. I would thus like to express my appreciation to the authors for having earnestly thought through my feedback, and for responding in kind. I have a few remaining comments which I personally still regard as important and essential so that the paper is presented as transparently as possible, but I do not suspect any would require much work to address.

1) Turnover of users and hashtags between slices

I appreciate the authors clarification regarding the turnover of nodes between slices, and the addition of Figure S2 to the Supplementary Material. Nevertheless, I still feel as if this detail could be too easily overlooked by the reader. In first instance, I believe that the sentence about slices early in the Results should be amended to state:

"Each slice is represented as a bipartite network with a fixed number of the most active actors *in that slice*..."

(As an aside, thoroughly reviewing and commenting about a paper without line numbers is a brutal experience. Since the authors clearly prepared their manuscript with LaTeX, it takes essentially no effort to use the lineno package, and the journal guidelines in no way preclude the presence of line numbering. Please consider this small improvement to make things substantially easier for future reviewers.)

In second instance, I feel like the authors should be more conservative when they talk about "network flexibility" which could give a false impression about what is and isn't changing over time. The authors might feel I'm being overly dramatic with this point, but from my perspective this seemingly minor detail fundamentally alters the way in which the authors empirical results should be interpreted. As I mentioned in my previous review, the (perfectly understandable) turnover between nodes implies that it isn't actually the *same* network that is showing flexibility in the face of an external shock. Instead, what the authors see in their data is that the most-active component of a *much* larger network shows characteristic shifts in its macroscopic properties. The potential implications are very different between what the authors study and what would occur in a situation in which the user sets and hashtag sets were constant throughout the time window. This detail does not render the authors' results uninteresting or false, and I'm not suggesting the authors conduct an analysis with all nodes (which is computationally infeasible) or an analysis with fixed nodes (though that would also be very cool!). But it absolutely changes how the dynamics should be conceptualised by the reader and where an inquisitive researcher might attempt to take the results in the future. As such, I feel like perfect clarity here is paramount.

2) Competition between users and hashtags

The authors' model includes parameters β_{ij}^U and β_{ij}^H that define within-guild competitive interactions. Mathematically, these terms are important in order to prevent nodes from growing unbounded (as shown across innumerable previous studies of mutualistic models). However, as noted in their response to the last round of review, the conceptual rationale for these terms is much murkier than the postulated rationale for the γ_{ik}^{UH} . If the authors have even a verbal argument for why these competitive effect could arise, I'd love to see it when they define the specifics of their model. If they do not, I would also appreciate a qualifier to the effect of "These competitive effects are important to obtain realistic node dynamics, but their underlying mechanistic basis remains an open question."

(Technical aside: I wish I would have noticed this previously, but shouldn't the denominators of

the functional responses in Eq. (4) include γ_{ik}^{UH} (the strength of the mutualism, which plays the role of an attack rate) and not θ_{ik}^{UH} (which is just 0 or 1 as defined by the adjacency matrix?)

3) "Expected" versus unexpected/unexplained transitions

I appreciate the new text regarding unexpected or unexplained transitions near the top of page 4. However, I have a really hard time buying the notion that these anecdotal observations belong in the Results. I suggest moving this to the Discussion, and perhaps concluding with the suggestion that distinguishing between expected and unexpected, remarkable and unremarkable, transitions in dynamic networks is an open and exiting question. Moreover, there may be instances when a transition *is* expected but isn't observed. This "inflexibility" of a network could also be something useful for people to explore going forward as there may be utility in both flexibility and antifragility (sensu Taleb).

4) Hashtag frequencies in Figure 4

I accept that I am probably being daft, but I can think of many different mathematical operations that could potentially correspond to "...hashtag frequencies are shifted to their initial value." If this just means the differences between their relative abundance at time t and their relative abundance at time 0, it may be clearest to just write this out in equation form and at least change the y-axis label to "relative hashtag freq."

Nature Communications - response to reviewers (III)

Manuscript Number: NCOMMS-20-18890B

Structural Flexibility in Online Communication Networks: an ecological approach

Reviewer #5

I feel that the revised version of the manuscript has clarified the vast majority of my concerns. I would thus like to express my appreciation to the authors for having earnestly thought through my feedback, and for responding in kind.

We thank the reviewer for the positive comments on the clarifications to their concerns, and on the value of the manuscript.

I have a few remaining comments which I personally still regard as important and essential so that the paper is presented as transparently as possible, but I do not suspect any would require much work to address.

Below we offer a point-by-point response to address the issues raised by the referee.

1) Turnover of users and hashtags between slices

I appreciate the authors' clarification regarding the turnover of nodes between slices, and the addition of Figure S2 to the Supplementary Material. Nevertheless, I still feel as if this detail could be too easily overlooked by the reader. In first instance, I believe that the sentence about slices early in the Results should be amended to state:

*"Each slice is represented as a bipartite network with a fixed number of the most active actors *in that slice*..."*

The referee is right, in the sense that the manuscript needs to be self-contained at least in the core aspects that might be confusing. We have followed the advice, and changed the sentence in the main text, lines 89-91, with an explicit reference to Figure S2 of the SM.

(As an aside, thoroughly reviewing and commenting about a paper without line numbers is a brutal experience. Since the authors clearly prepared their manuscript with LaTeX, it takes essentially no effort to use the lineno package, and the journal guidelines in no way preclude the presence of line numbering. Please consider this small improvement to make things substantially easier for future reviewers.)

We apologise for overlooking this detail. The new version of the manuscript includes line numbers, and our modifications are identified *via* line numbers in this letter (besides colour highlights).

*In second instance, I feel like the authors should be more conservative when they talk about "network flexibility" which could give a false impression about what is and isn't changing over time. The authors might feel I'm being overly dramatic with this point, but from my perspective this seemingly minor detail fundamentally alters the way in which the authors empirical results should be interpreted. As I mentioned in my previous review, the (perfectly understandable) turnover between nodes implies that it isn't actually the *same* network that is showing flexibility in the face of an external shock. Instead, what the authors see in their data is that the most-active component of a *much* larger network shows characteristic shifts in its macroscopic properties. The potential implications are very different between what the authors study and what would occur in a situation in which the user sets and hashtag sets were constant throughout the time window. This detail does not render the authors' results uninteresting or false, and I'm not suggesting the authors conduct an analysis with all nodes (which is computationally infeasible) or an analysis with fixed nodes (though that would also be very cool!). But it absolutely changes how the dynamics should be conceptualised by the reader and where an inquisitive researcher might attempt to take the results in the future. As such, I feel like perfect clarity here is paramount.*

We see the referee's point, and we agree that this is a subtle issue that may trigger criticism: considering some discussions in the last reviewing round, we can agree that the system has a temporal continuity (that is, an internal coherence of the topic being discussed through time), but the networked slices of the system are not identical through time. This is, briefly speaking, the dominant point of view in time-resolved studies of mutualistic communities (ref. [33] in the main text, refs. [11,19,23] in the Supplementary Material).

For this reason, we have changed the expression "network flexibility" for "system flexibility". It turns out that "network flexibility" was used in the text only twice (one of them in the title), so we have changed it accordingly.

2) Competition between users and hashtags

The authors' model includes parameters β_{ij}^U and β_{ij}^H that define within-guild competitive interactions. Mathematically, these terms are important in order to prevent nodes from growing unbounded (as shown across innumerable previous studies of mutualistic models). However, as noted in their response to the last round of review, the conceptual rationale for these terms is much murkier than the postulated rationale for the γ_{ik}^{UH} . If the authors have even a verbal argument for why these competitive effect could arise, I'd love to see it when they define the specifics of their model. If they do not, I would also appreciate a qualifier to the effect of "These competitive effects are important to obtain realistic node dynamics, but their underlying mechanistic basis remains an open question."

We thank the referee for providing the opportunity to clarify this point further. The competitive interactions were thoroughly discussed in the last round, and we are sorry to see that our

arguments and references sound (or read) murky. Our inclusion of these interactions does not respond to a mathematical convenience, but to the need to model an actual driver of the system. On the memes or hashtags side, lines 23-31 argue (admittedly with brevity) how competitive interactions emerge in language systems. On the users side, competition for attention is supported by several works in four different fields: the complex network literature (refs. [11,13-15]), the computer science literature (refs. [6,8]), and the cognitive and neuroscience literature (refs. [7,9,12]). In the Introduction, where those arguments were explicitly outlined, we have rephrased part of the text (line 21), and relocated some references. Also, we have highlighted the nature of these competitive interactions again in the model outline (lines 153-155), and in the Methods summary (where β is explicitly mentioned, lines 376-377).

(Technical aside: I wish I would have noticed this previously, but shouldn't the denominators of the functional responses in Eq. (4) include γ_{ik}^{UH} (the strength of the mutualism, which plays the role of an attack rate) and not θ_{ik}^{UH} (which is just 0 or 1 as defined by the adjacency matrix)?)

We have reviewed the Methods Summary (subsection Modelling Framework) and realised that there was a typo in the expression defining γ_{ik}^{UH} --or, to be more precise, there was an important piece missing: the correct expression is

$$\gamma_{ik}^{UH} \propto \Omega_m \cdot G_{ik}^{UH} \cdot \theta_{ik}^{UH}$$

where θ_{ik}^{UH} had not been included explicitly in the main text (this has been corrected, line 379). We suspect that this may have misled the referee, prompting their comment, because only now it becomes evident that γ_{ik}^{UH} has the same shape as θ_{ik}^{UH} . Since

$$\gamma_{ik}^{UH} \leq \theta_{ik}^{UH}, \forall (i, k)$$

one may interpret the denominator as the best possible mutualistic scenario, and thus the maximum attack rate.

In any case, we acknowledge that the literature is diverse regarding the particular formulation of Lotka-Volterra mutualistic dynamics, and most probably we will not settle the issue. Here, regarding the saturation term, we literally follow ref. [31] (Cai, W., et al. Nat. Comm. 2020), which we consider a moderate assumption compared to the denominator of this term in Suweis et al. (Nature 2013; ref. [29]), where $\theta_{ij} = 1$ for all (i, j) pairs (i.e. a complete bipartite network of 1s).

3) "Expected" versus unexpected/unexplained transitions

*I appreciate the new text regarding unexpected or unexplained transitions near the top of page 4. However, I have a really hard time buying the notion that these anecdotal observations belong in the Results. I suggest moving this to the Discussion, and perhaps concluding with the suggestion that distinguishing between expected and unexpected, remarkable and unremarkable, transitions in dynamic networks is an open and exciting question. Moreover, there may be instances when a transition *is* expected but isn't observed. This "inflexibility" of a network could also be something useful for people to explore going forward as there may be utility in both flexibility and antifragility (sensu Taleb).*

We share the referee's opinion on this point --the Results section is already too dense, and the inclusion of this particular aspect does not help. Thus we have moved this item to the Discussion, now lines 296-300. The new paragraph includes explicitly some of the referee's suggestions, and for this reason we have added a note in the Acknowledgements section.

4) Hashtag frequencies in Figure 4

I accept that I am probably being daft, but I can think of many different mathematical operations that could potentially correspond to "...hashtag frequencies are shifted to their initial value." If this just means the differences between their relative abundance at time t and their relative abundance at time 0 , it may be clearest to just write this out in equation form and at least change the y-axis label to "relative hashtag freq."

Hashtag frequencies in Figure 4 are presented in absolute value, shifted to the absolute frequency observed at the beginning of the observation. We agree that this remains much clearer adding an explicit expression, $\hat{f}_h = f_h(t) - f_h(0)$ (the notation \hat{f} is used for coherence with Figs. 1 and 3). We have modified the caption and associated text following the referee's advice, to avoid ambiguities (lines 236-238).

Reviewers' Comments:

Reviewer #5:

Remarks to the Author:

I again appreciate the authors for taking their time to consider my previous review. I am very happy with the present state of the manuscript.